# Ca²⁺ entry into neurons is facilitated by cooperative gating of clustered Ca$_V$1.3 channels

**Claudia M Moreno[1], Rose E Dixon[2], Sendoa Tajada[1], Can Yuan[2], Ximena Opitz-Araya[2], Marc D Binder[2]\*, Luis F Santana[1]\***

[1]Department of Physiology and Membrane Biology, University of California, Davis, United States; [2]Department of Physiology and Biophysics, University of Washington, Seattle, United States

**Abstract** Ca$_V$1.3 channels regulate excitability in many neurons. As is the case for all voltage-gated channels, it is widely assumed that individual Ca$_V$1.3 channels behave independently with respect to voltage-activation, open probability, and facilitation. Here, we report the results of super-resolution imaging, optogenetic, and electrophysiological measurements that refute this long-held view. We found that the short channel isoform (Ca$_V$1.3$_S$), but not the long (Ca$_V$1.3$_L$), associates in functional clusters of two or more channels that open cooperatively, facilitating Ca²⁺ influx. Ca$_V$1.3$_S$ channels are coupled via a C-terminus-to-C-terminus interaction that requires binding of the incoming Ca²⁺ to calmodulin (CaM) and subsequent binding of CaM to the pre-IQ domain of the channels. Physically-coupled channels facilitate Ca²⁺ currents as a consequence of their higher open probabilities, leading to increased firing rates in rat hippocampal neurons. We propose that cooperative gating of Ca$_V$1.3$_S$ channels represents a mechanism for the regulation of Ca²⁺ signaling and electrical activity.

**\*For correspondence:** mdbinder@
uw.edu (MDB); lfsantana@ucdavis.
edu (LFS)

**Competing interests:** The authors declare that no competing interests exist.

## Introduction

Ca$_V$1.3 channels are widely expressed in neurons throughout the brain and spinal cord (*Tan et al., 2011*), where they serve a number of critical functions including the modulation of resting potentials, the amplification of synaptic currents and the generation and shaping of repetitive firing (*Guzman et al., 2009*; *Olson et al., 2005*; *Striessnig et al., 2006*). These channels are dihydropyridine-sensitive L-type Ca²⁺ channels composed of a pore-forming Ca$_V$1.3$\alpha_1$ subunit and accessory $\beta$ and $\alpha_2$-$\delta$ subunits. The carboxy-terminus (C-terminus) of the $\alpha_{1D}$ subunit is structurally complex, containing an EF hand domain as well as pre-IQ and IQ domains to which the Ca²⁺-binding protein calmodulin (CaM) binds (*Ben-Johny and Yue, 2014*).

Alternative splicing results in the expression of 'long' and 'short' Ca$_V$1.3 channel isoforms that differ in the length of the C-terminus (*Singh et al., 2008*). The splice variant 42A (Ca$_V$1.3$_S$) has a short C-terminus of 183 amino acids long compared to the 695 amino acids of the long isoform (Ca$_V$1.3$_L$). The Ca$_V$1.3$_S$ channels lack the so-called C-terminal modulatory domain (CTM), comprised of proximal (PCRD) and distal (DCRD) regulatory domains that block CaM binding to the IQ domain (*Figures 1A and B*, left). As a consequence, Ca$_V$1.3$_S$ channels activate at lower voltages, have a higher open probability, and inactivate faster than Ca$_V$1.3$_L$ channels (*Bock et al., 2011*; *Singh et al., 2008*; *Tan et al., 2011*).

Ca$_V$1.3 channels carry about 20% of the L-type calcium current in hippocampal neurons (*Moosmang et al., 2005*). As Ca$_V$1.3$_S$ channels activate at low voltages, they are a particular good candidate for underlying the sustained the neuronal firing and the persistent low- voltage-activated

**eLife digest** The electrical charge inside a cell is different from that outside of the cell. Neurons rely on this difference to send signals via electrical impulses. This process involves ions moving across the neuron's membrane through proteins called ion channels. Ca$_v$1.3 channels are ion channels that open when the membrane's electrical charge changes to allow positively charged calcium ions into the cell. This generates an electrical current that enables neurons in the brain to produce repetitive impulses.

Calcium ions entering through a Ca$_v$1.3 channel can encourage the channel to allow in even more calcium ions. A closely related channel called Ca$_v$1.2, which is essential to the activity of heart muscle, behaves in a similar way. Researchers have recently found that Ca$_v$1.2 channels are arranged in clusters in the membrane and that adjacent channels interact to allow more calcium ions through the channels. This was an unexpected finding because it had long been thought that all ion channels acted independently.

Moreno et al. have now used a range of different approaches to investigate the behavior of one form of the Ca$_v$1.3 channel, called Ca$_V$1.3$_S$, in human cells and in neurons from rat brains. Initial experiments confirmed that calcium ions stimulated these channels to open in a coordinated way and to allow in more calcium. High-resolution microscopy then revealed that the Ca$_V$1.3$_S$ channels do form clusters in the cell membrane.

Moreno et al. went on to demonstrate that this simultaneous opening of Ca$_V$1.3$_S$ channels first requires a protein called calmodulin to bind calcium inside the cell. Next, the calcium-calmodulin complex associates with the parts of the channels that are also inside the cell. Further experiments showed that coupling the Ca$_v$1.3 channels together allows them to cooperate, and makes them more likely to be open and generate bigger calcium flows, and allowed neurons to send electrical signals more frequently. Future challenges include investigating how the clusters of Ca$_v$1.3 channels are established and maintained, and determining how the channels' cooperation plays a role in both healthy and diseased states.

current observed in CA3 neurons (*Avery and Johnston, 1996*). In fact, Ca$_V$1.3 channels support spontaneous firing of dopaminergic neurons in the substantia nigra and mid-spiny striatal neurons (*Guzman et al., 2009*; *Olson et al., 2005*).

The function of Ca$_V$1.3 channels is tightly regulated by changes in intracellular Ca$^{2+}$ ([Ca$^{2+}$]$_i$). The opening of Ca$_V$1.3 channels causes a local increase in [Ca$^{2+}$]$_i$ that induces two opposing regulatory mechanisms: Ca$^{2+}$-dependent inactivation (CDI) and Ca$^{2+}$-dependent facilitation (CDF) (*Ben-Johny and Yue, 2014*). CDF manifests as an increase in the magnitude of the Ca$_V$1.3 current ($I_{Ca}$) with repetitive activation. In neurons, CDF can induce a persistent $I_{Ca}$ that increases firing rate and may even lead to self-sustained firing (*Fransen et al., 2006*; *Major and Tank, 2004*; *Sheffield et al., 2013*). It has been proposed that CDF of the Ca$_V$1.3 channel depends on Ca$^{2+}$/CaM-dependent kinase II (CaMKII)-mediated phosphorylation, as has also been proposed for the closely related Ca$_V$1.2 channel (*Hudmon et al., 2005*; *Xiao et al., 1994*; *Yuan and Bers, 1994*). This phosphorylation requires the presence of a second protein, densin, which binds to the PDZ domain located in the most distal part of the C-terminus of the channel (*Jenkins et al., 2010*). Because Ca$_V$1.3$_S$ lacks that PDZ domain, CaMKII-mediated phosphorylation is unlikely to be responsible for CDF in Ca$_V$1.3$_S$ channels. Thus, the mechanisms underlying the CDF of the widely expressed Ca$_V$1.3$_S$ channel have not yet been resolved.

Two recent studies by Dixon et al. (*Dixon et al., 2012*; *2015*) have suggested the tantalizing hypothesis that Ca$^{2+}$-induced interactions between the C-termini of neighboring Ca$_V$1.2 channels facilitates Ca$^{2+}$ influx by increasing the activity of adjoined channels in cardiac muscle. At present, however, whether this physical and functional coupling of Ca$_V$1.2 channels is a common mechanism for the control of Ca$^{2+}$ influx via voltage-gated Ca$^{2+}$ channel function, including Ca$_V$1.3 channels is unknown. Furthermore, the possibility that cooperative Ca$_V$1.3 channel gating regulates neuronal excitability is also unclear.

In the present study, using electrophysiological, optogenetic, and super-resolution imaging approaches, we discovered that $Ca_V1.3_S$ channels form functional clusters of two or more channels along the surface membrane of hippocampal neurons. Clustered $Ca_V1.3_S$ channels undergo $Ca^{2+}$-induced physical interactions that increase the activity of adjoined channels, facilitate $Ca^{2+}$ currents and thereby increase firing rates in hippocampal neurons. We propose that cooperative gating of $Ca_V1.3_S$ channels is a new general mechanism for the amplification of $Ca^{2+}$ signals in excitable cells.

## Results

### A $Ca^{2+}$-dependent mechanism mediates facilitation of $Ca_V1.3_S$, but not $Ca_V1.3_L$, channels

Because $Ca_V1.3$ channels are alternatively spliced, we first sought to determine whether $Ca_V1.3_S$ and $Ca_V1.3_L$ channels are differentially regulated by $[Ca^{2+}]_i$. Macroscopic currents were recorded from tsA-201 cells expressing either $Ca_V1.3_S$ or $Ca_V1.3_L$ channels in the presence of 2 mM $Ba^{2+}$ or 2 mM $Ca^{2+}$. Currents were activated by a depolarizing pulse (300 ms) from a holding potential of -80 mV to -10 mV. With $Ba^{2+}$ in the external solution, membrane depolarization induced large $Ca_V1.3_L$ currents that inactivated slowly (*Figure 1A*, center). Switching to a perfusion solution containing $Ca^{2+}$ decreased the amplitude of $Ca_V1.3_L$ currents by nearly 40% and increased the rate of inactivation. Like $Ca_V1.3_L$ currents, $Ca_V1.3_S$ currents inactivated faster when $Ca^{2+}$ was used as a charge carrier however, in agreement with previous reports (*Bock et al., 2011*; *Singh et al., 2008*), we observed more pronounced CDI (defined as the difference between inactivation of $I_{Ba}$ and $I_{Ca}$) in $Ca_V1.3_S$ channels compared to the $Ca_V1.3_L$ variant (*Figure 1B*, right versus *Figure 1A*, right). As discussed above, this difference in the magnitude of CDI has been attributed to the lack of the CTM domain in $Ca_V1.3_S$ channels. Curiously, the amplitude of $Ca_V1.3_S$ currents decreased to a lesser extent (only about 15% at -10 mV) upon changing the external solution from $Ba^{2+}$ to $Ca^{2+}$ (*Figure 1B*, center).

We investigated whether differences in the amplitude of elementary $Ca_V1.3_L$ and $Ca_V1.3_S$ channel currents could, at least in part, account for these disparities in macroscopic $Ca^{2+}$ and $Ba^{2+}$ currents. Single $Ca_V1.3_L$ and $Ca_V1.3_S$ channel currents were recorded from cell-attached patches with pipettes containing 20 mM $Ca^{2+}$ or $Ba^{2+}$. With $Ca^{2+}$ as the charge carrier, the amplitudes of elementary $Ca_V1.3_S$ and $Ca_V1.3_L$ channel currents were similar. For example, at -30 mV they were -0.48 ± 0.07 and -0.49 ± 0.01 pA for $Ca_V1.3_L$ and $Ca_V1.3_S$ channels, respectively (*Figures 1C–F*). These values are in accordance with the unitary currents reported by Guia et al. for cardiac L-type channels using $Ca^{2+}$ as charge carrier (*Guia et al., 2001*). Ensemble averages revealed currents that activated quickly and then inactivated likely due to $Ca^{2+}$ and voltage-dependent mechanisms (*Figure 1G*). Furthermore, as is the case for other $Ca_V1$ channels, the single channel currents produced during the opening of $Ca_V1.3_L$ and $Ca_V1.3_S$ channels were both larger with $Ba^{2+}$ as the charge carrier than with $Ca^{2+}$ (-1.14 ± 0.02 pA and -1.10 ± 0.02 pA for $Ca_V1.3_L$ and $Ca_V1.3_S$ at -30 mV, respectively), but not significantly different from each other (*Figure 1—figure supplement 1*). The amplitude of the unitary currents with $Ba^{2+}$ is also in accordance with previously reported values for these channels (*Bock et al., 2011*). These data suggest that the difference observed in the macroscopic currents between the $Ca_V1.3_L$ and $Ca_V1.3_S$ channels is not due to differences in unitary currents.

We then tested the hypothesis that $Ca^{2+}$ enhances the activation of $Ca_V1.3_S$, but not $Ca_V1.3_L$ channels by increasing the channel activity ($NP_o$). We performed a quantitative analysis of $NP_o$. The whole-cell current (I) is given by the equation $I = i*N*P_o$, where $i$ is the amplitude of the elementary current, $N$ is the number of functional channels, and $P_o$ is the channel open probability. Since elementary $Ca_V1.3_L$ and $Ca_V1.3_S$ currents are larger when $Ba^{2+}$ rather than $Ca^{2+}$ is the charge carrier, for $I_{Ca}$ to be similar to $I_{Ba}$, the $NP_o$ of $Ca_V1.3_S$ must be higher in the presence of $Ca^{2+}$ than $Ba^{2+}$. We estimated $NP_o$ with $Ca^{2+}$ and $Ba^{2+}$ as charge carriers by dividing the amplitude of whole-cell $Ca_V1.3_S$ and $Ca_V1.3_L$ currents at -10 mV, by the values of unitary current. Our analysis relies on the assumption that the relatively larger peak of $Ca_V1.3_S$ currents in the presence of $Ca^{2+}$ was not due to faster activation kinetic of this channel than that of $Ca_V1.3_L$ channels. This assumption is reasonable, as we found no significant difference in the activation time constants for $Ca_V1.3_L$ and $Ca_V1.3_S$ currents that were 1.60 ± 0.22 and 1.17 ± 0.051 ms (-10 mV, n = 6 for each channel, p = 0.112), respectively. We used elementary currents values recorded with 2 mM $Ca^{2+}$ (-0.16 pA) and $Ba^{2+}$ (-0.24 pA) at -10 mV (*Guia et al., 2001*). We found that $Ca^{2+}$ ions entering the cell through the

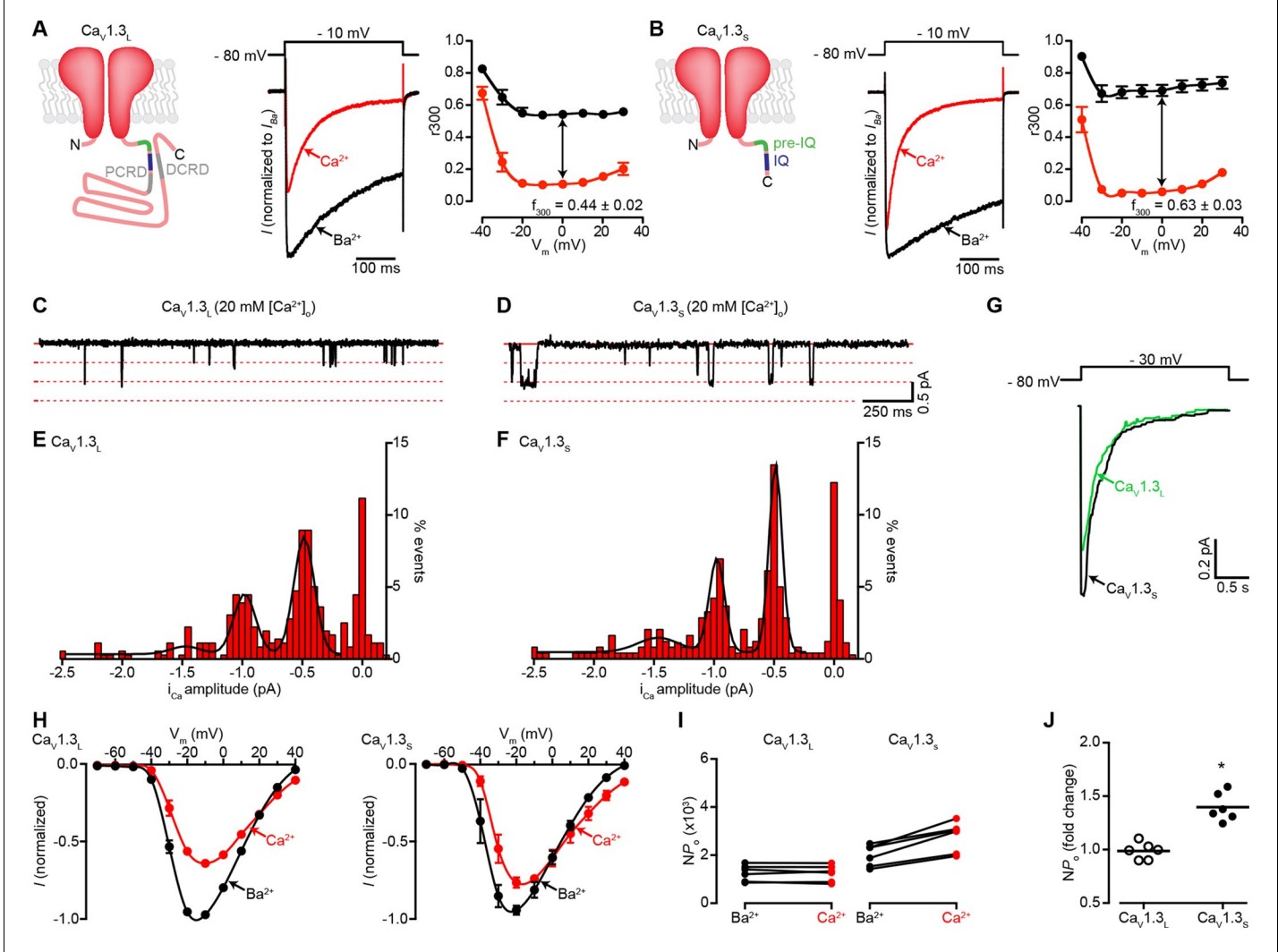

**Figure 1.** $Ca^{2+}$ enhances the activity of $Ca_V1.3_S$, but not $Ca_V1.3_L$, channels. (A) *Left*: Schematic of $Ca_V1.3_L$ channel splice variant, depicting the domains important for $Ca^{2+}$-mediated regulation: pre-IQ (green), IQ (blue), proximal and distal C-terminal regulatory domains (PCRD, DCRD, gray). *Middle*: Representative $I_{Ca}$ and $I_{Ba}$ of $Ca_V1.3_L$ channels expressed in tsA-201 cells. Currents were evoked by a 300-ms depolarization from holding potential of -80 mV to a test potential of -10 mV, with 2 mM $Ba^{2+}$ (black) or 2 mM $Ca^{2+}$ (red) as the charge carrier in the same cell. *Right*: Voltage dependence of $Ca_V1.3_L$ channel CDI. r300 is the fraction of $I_{Ca}$ or $I_{Ba}$ remaining after 300 ms. $f_{300}$ is the difference between $I_{Ca}$ and $I_{Ba}$ r300 values at 0 mV. (B) *Left*: Schematic of $Ca_V1.3_S$ channel splice variant. *Middle*: Representative $I_{Ca}$ and $I_{Ba}$ of $Ca_V1.3_S$ channels. Right: Voltage dependence of $Ca_V1.3_S$ channel CDI, format as in (A). $I_{Ca}$ is presented as normalized to $I_{Ba}$, currents analyzed for these experiments were in a range between 200 and 900 pA. (C and D) Representative $i_{Ca}$ single channel recordings from $Ca_V1.3_L$ (C) and $Ca_V1.3_S$ channels (D) expressed in tsA-201 cells during step depolarizations from -80 to -30 mV. (E and F) all-points $i_{Ca}$ amplitude histograms for $Ca_V1.3_L$ (E) and $Ca_V1.3_S$ channels (F), the black line is the best fit to the data with a multi-Gaussian function with a quantal unit value of -0.48 ± 0.07 for $Ca_V1.3_L$ and -0.49 ± 0.01 pA for $Ca_V1.3_S$ channels, respectively (constructed from *n* = 6 cells each). Single channel recordings were also performed using $Ba^{2+}$ as the charge carrier for both channels (see *Figure 1—figure supplement 1*). (G) Ensemble average single-channel currents from multiple sweeps. (H) Current-voltage relationships of $Ca_V1.3_L$ currents (left) and $Ca_V1.3_S$ currents (right) in the presence of 2 mM $Ca^{2+}$ (red) or $Ba^{2+}$ (black) as the charge carrier. Data were normalized to the maximum current in the presence of $Ba^{2+}$. Symbols are averages of 7 cells ± SEM. (I) Scatter plots of $NP_o$ (at -10 mV) of $Ca_V1.3_L$ (left) and $Ca_V1.3_S$ (right) channels in the presence of $Ba^{2+}$ and $Ca^{2+}$. (J) Change in $NP_o$ for $Ca_V1.3_L$ and $Ca_V1.3_S$ channels for currents recorded in the presence of $Ca^{2+}$ and then $Ba^{2+}$. The horizontal bar shows the mean value (*p < 0.001).

The following figure supplement is available for figure 1:

**Figure supplement 1.** Single-channel recordings of $i_{Ba}$ for $Ca_V1.3_S$ and $Ca_V1.3_L$ channels.

channels increased the $NP_o$ nearly 1.5-fold for $Ca_V1.3_S$ channels, but not at all for $Ca_V1.3_L$ channels (*Figures 1I and J*). Thus, assuming that the number of functional $Ca_V1.3$ channels ($N$) in the membrane remained constant, a reasonable assumption given the short time lapse (~2 min) between recording $I_{Ba}$ and $I_{Ca}$ from the same cell, these data suggest that a $Ca^{2+}$-dependent mechanism enhances inward $Ca^{2+}$ currents by increasing the $P_o$ of $Ca_V1.3_S$ channels, but not that of $Ca_V1.3_L$ channels.

## $Ca_V1.3_S$ channels gate cooperatively, increasing $Ca^{2+}$ influx

One possible explanation for the $Ca^{2+}$-influx–dependent increase in the $P_o$ of $Ca_V1.3_S$ channels is cooperative gating among channels in small clusters, as we have previously reported for $Ca_V1.2$ channels in cardiomyocytes and smooth muscle cells (*Navedo et al., 2010*). To test this possibility, we made optical recordings of individual $Ca_V1.3_S$-mediated $Ca^{2+}$ influx events (called 'sparklets') in $Ca_V1.3_L$ and $Ca_V1.3_S$-expressing tsA-201 cells loaded with 200 µM Rhod-2 using total internal reflection fluorescence (TIRF) microscopy. $Ca_V1.3_S$ sparklets were recorded at a membrane potential of -80 mV in the presence of 20 mM external $Ca^{2+}$; 10 mM EGTA was included in the patch pipette to confine the $[Ca^{2+}]_i$ signal to within ~1 µm of the point of $Ca^{2+}$ entry (*Zenisek et al., 2003*). A quantal analysis of $Ca_V1.3_S$ and $Ca_V1.3_L$ sparklets revealed the presence of single-level (elementary) events with a mean amplitude of ~40 nM for both channels, in agreement with our previous study (*Navedo et al., 2007*). Interestingly, consistent with our single channel data, we found that multi-quantal sparklets, which presumably result from the simultaneous opening of several $Ca_V1.3$ channels, were more commonly observed in cells expressing $Ca_V1.3_S$ than $Ca_V1.3_L$ channels (*Figure 2A*). To determine whether channels in a cluster opened cooperatively or independently, we calculated the coupling coefficient (κ) among channels within a $Ca_V1.3_S$ and $Ca_V1.3_L$ sparklet site by applying a coupled Markov-chain model (*Chung and Kennedy, 1996*). Channels with κ > 0.1 were considered coupled (*Navedo et al., 2010*). Using this approach, we found that the average κ value for $Ca_V1.3_S$ and $Ca_V1.3_L$ channels was 0.21 ± 0.05 (n = 15) and 0.08 ± 0.04 (n = 12), respectively (*Figure 2B*). These results support the hypothesis that $Ca_V1.3_S$ channels are more likely to undergo cooperative gating, generating persistent and greater $Ca^{2+}$ influx than $Ca_V1.3_L$ channels.

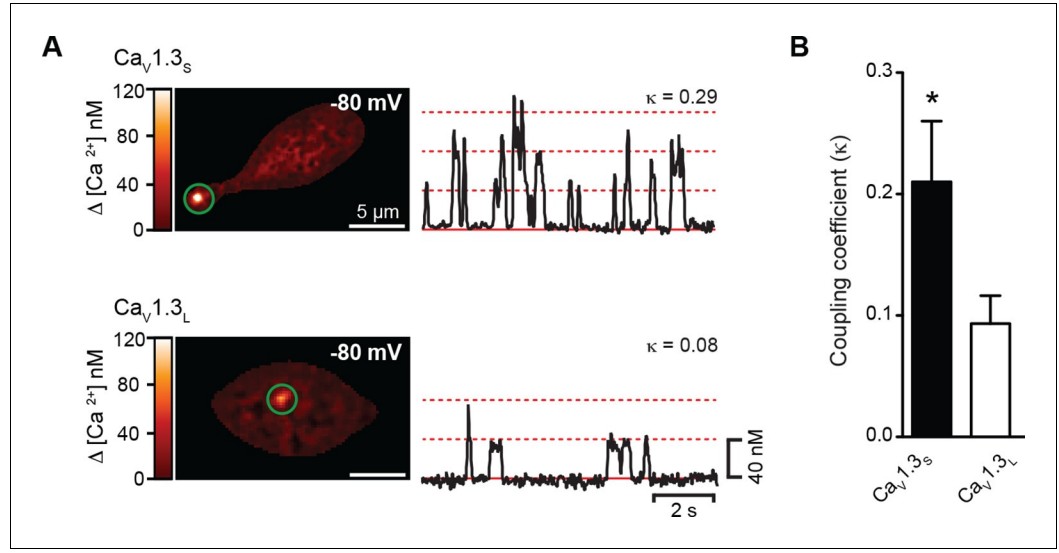

**Figure 2.** $Ca_V1.3_S$ but not $Ca_V1.3_L$ channels gate cooperatively to increase $Ca^{2+}$ influx. (**A**) TIRF images of spontaneous $Ca_V1.3_S$ (top) and $Ca_V1.3_L$ sparklets (bottom) at a holding potential of -80 mV in tsA-201 cells expressing the respective channels. Traces at the right show the time course of $[Ca^{2+}]_i$ in the sites marked by the green circles. The dotted red lines show the amplitudes of 1 to 3 quantal levels. The coupling coefficient (κ) is shown above each trace. (**B**) Bar chart showing the coupling coefficient for the $Ca_V1.3_S$ and $Ca_V1.3_L$ sparklets sites. Bars are averages of 5 cells ± SEM (*p<0.05).

## $Ca_V1.3$ channels in hippocampal neurons aggregate in dense clusters

If the signal for cooperative $Ca_V1.3_S$ channel gating is a local increase in $[Ca^{2+}]_i$, these channels must be in close proximity to one another. To test this hypothesis, we examined the spatial organization of endogenous $Ca_V1.3$ channels in hippocampal neurons using super-resolution localization microscopy (*Figure 3A–D*). Hippocampal neurons were immunostained against $Ca_V1.3$ using an antibody kindly provided by Dr. William Catterall and Dr. Ruth Westenbroek. This analysis showed that $Ca_V1.3$ channels form clusters occupying an average area of $3660 \pm 80$ nm$^2$ (n = 5). The antibody used in this study has been shown to not recognize the corresponding sequence of the closely related $Ca_V1.2$ $\alpha$ subunit in both, transfected cells and hippocampal tissue (*Hell et al., 1993*; *2013*). However, we were not able to test the specificity of the antibody on $Ca_V1.3$-KO neurons and thus, pursued our analyses of channel clustering using a heterologous expression of $Ca_V1.3$ channels in tsA-201 cells.

The specificity of the antibody in tsA-201 cells was tested by immunostaining untransfected and $Ca_V1.3_S$-transfected cells. No evidence of staining was observed in the untransfected cells (*Figure 3—figure supplement 1A*, n = 4). As our antibody cannot distinguish between $Ca_V1.3$ channels isoforms, we expressed $Ca_V1.3_L$ or $Ca_V1.3_S$ channels separately and found that both channel subtypes form clusters of similar size (*Figure 3—figure supplement 1B–F*). The mean areas of $Ca_V1.3_L$ and $Ca_V1.3_S$ channel clusters were $2543 \pm 50$ nm$^2$ and $2119 \pm 73$ nm$^2$, respectively (*Figure 3—figure supplement 1G*, n = 7). GSD images were acquired in the TIRF focal plane with a penetration depth of 130 nm.

To determine the number of channels within $Ca_V1.3$ channel clusters, we used step-photobleaching (*Ulbrich and Isacoff, 2007*), of expressed green fluorescent protein (GFP)-fused $Ca_V1.3_S$ channels in hippocampal neurons (*Figure 3E and F*) and tsA-201 cells (*Figure 3—figure supplement 1G and H*). The rationale for using only $Ca_V1.3_S$-GFP channels in these studies is twofold. *First*, $Ca_V1.3_S$ and not $Ca_V1.3_L$ channels have the ability to undergo coupled activation. *Second*, the size of $Ca_V1.3_L$ and $Ca_V1.3_S$ puncta (at least in tsA-201 cells) is similar. Thus, $Ca_V1.3_L$ and $Ca_V1.3_S$ clusters are likely composed of the same number of channels. We identified and excited single $Ca_V1.3_S$ channel clusters in hippocampal neurons and tsA-201 cells using TIRF microscopy with a penetration depth of 130 nm. Although some intracellular signal could be collected, the use of TIRF restricts the signal mainly to the plasmalemmal fraction of the channels. After continuous photobleaching we counted the number of step-wise decreases in fluorescence intensity of $Ca_V1.3_S$-GFP clusters. The majority (65%) of $Ca_V1.3_S$ clusters underwent at least five stepwise decreases in fluorescence, with the remaining clusters showing fewer steps. A single photobleaching step was observed in only 4% of the $Ca_V1.3_S$ clusters. The mean number of $Ca_V1.3_S$ channels per cluster determined through sequential photobleaching was $8 \pm 1$ (n = 1105 clusters, 18 cells) in hippocampal neurons, this number is consistent with the average cluster area calculated from our super-resolution data and is similar to that reported in two different studies on the L-type channel distribution in hippocampal neurons using immunogold-labeling with electron microscopy and high-resolution immunofluorescence techniques (*Leitch et al., 2009*; *Obermair et al., 2004*). Step-photobleaching in tsA-201 cells revealed an average of $5 \pm 1$ $Ca_V1.3_S$ channels per cluster (n = 585 clusters, 11 cells), consistent also with our super-resolution cluster area measurements (*Figure 3—figure supplement 2*).

Collectively, these results support our working hypothesis that multi-quantal events detected by imaging $Ca^{2+}$ influx represent the simultaneous activity of multiple $Ca_V1.3_S$ channels in a membrane microdomain. Furthermore, our data suggest that although channel clustering may be necessary for functional coupling of adjacent $Ca_V1.3_S$ channels, physical clustering alone is not sufficient to induce functional coupling in $Ca_V1.3_L$ channels.

## Physical interaction between $Ca_V1.3_S$ channels induces $I_{Ca}$ facilitation

To determine whether a physical interaction between $Ca_V1.3_S$ channels induces $I_{Ca}$ facilitation, we fused the C-terminus of $Ca_V1.3_S$ and $Ca_V1.3_L$ channels to an optogenetic light-induced dimerization system based on CIBN and CRY2 proteins (*Kennedy et al., 2010*). Blue-light illumination (488 nm), promotes CIBN and CRY2 fusion, which forces the C-termini of the attached channels to interact (*Figure 4A*).

We transfected tsA-201 cells with $Ca_V1.3_S$-CIBN and $Ca_V1.3_S$-CRY2 or $Ca_V1.3_L$-CIBN and $Ca_V1.3_L$-CRY2 channels and measured $I_{Ca}$ in response to a series of depolarizing voltage steps before and

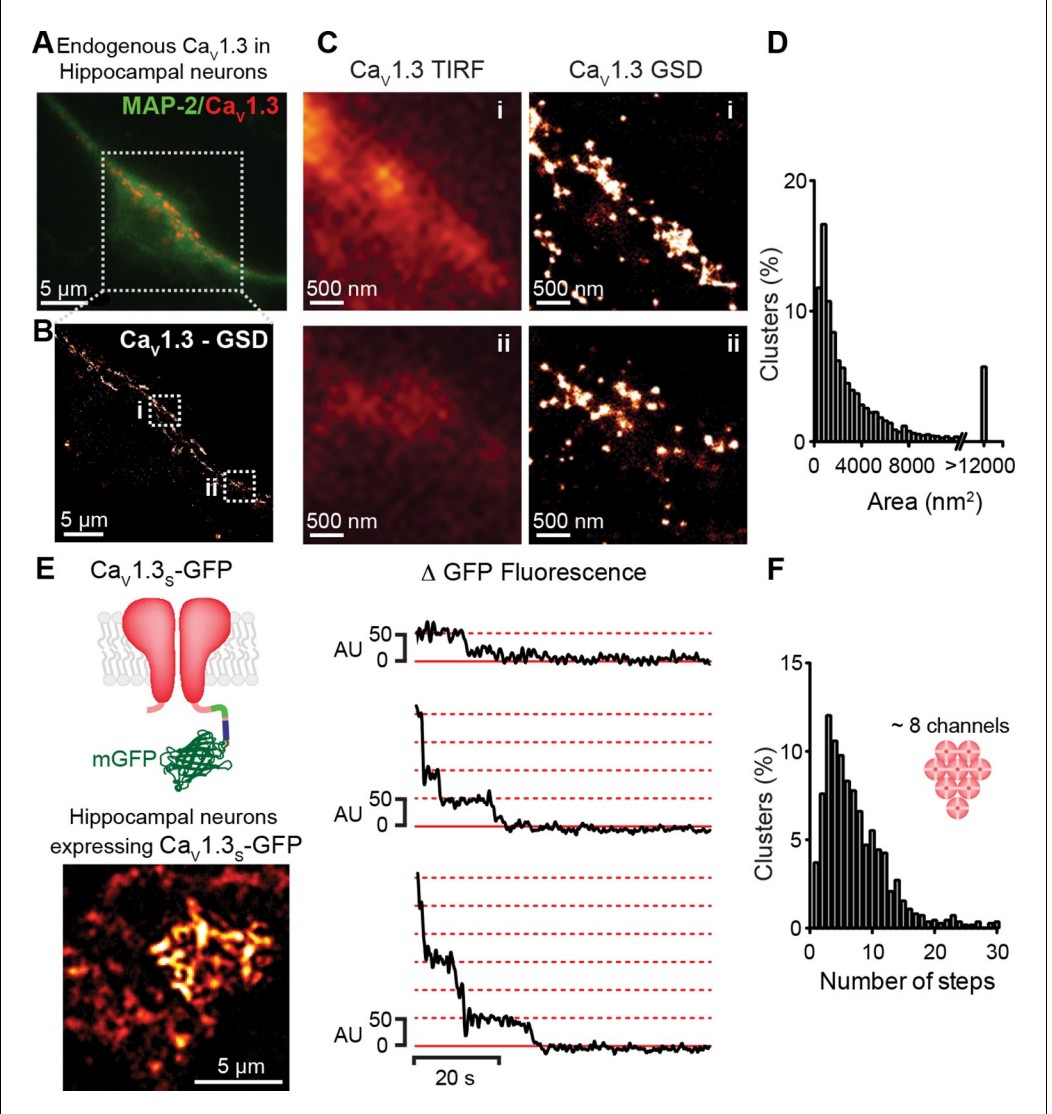

**Figure 3.** $Ca_V1.3$ channels assemble into clusters in the plasma membrane of cultured hippocampal neurons. (**A**) Wide-field image of a representative cultured hippocampal neuron immunostained for $Ca_V1.3$ channels (red) and the neuronal marker microtubule-associated protein 2 (MAP2; green). (**B**) Super-resolution (GSD) image of $Ca_V1.3$ channels in the outlined region in (**A**). (**C**) Comparison of conventional (TIRF, *left*) and super-resolution (GSD, *right*) images of $Ca_V1.3$ clusters in zones i and ii outlined in (**B**). (**D**) Frequency distribution of the area of $Ca_V1.3$ channel clusters (n = 5320 clusters from 5 cells). (**E**) TIRF image of $Ca_V1.3_S$-mGFP channels expressed in cultured hippocampal neurons (*left*). Examples of sequential photobleaching steps for three different clusters (*right*). (**F**) Frequency distribution of $Ca_V1.3_S$ cluster bleaching steps (n = 1105 clusters from 18 cells). Clustering of $Ca_V1.3_S$ and $Ca_V1.3_L$ channels was tested in tsA-201 cells expressing the respective isoform (See also *Figure 3—figure supplement 1*).

The following figure supplements are available for figure 3:

**Figure supplement 1.** $Ca_V1.3_S$ and $Ca_V1.3_L$ channels form clusters in tsA-201 cells.

**Figure supplement 2.** $Ca_V1.3_S$ organize in clusters of ~5 channels in tsA-201 cells.

after a 30-s exposure to blue light (488 nm) (*Figure 4B*). As shown in *Figure 4C*, in cells expressing $Ca_V1.3_S$-CIBN and $Ca_V1.3_S$-CRY2 channels, $I_{Ca}$ amplitude increased by 35% (n = 6, p<0.001) after illumination, whereas in cells expressing $Ca_V1.3_L$-CIBN and $Ca_V1.3_L$-CRY2 channels, there was no

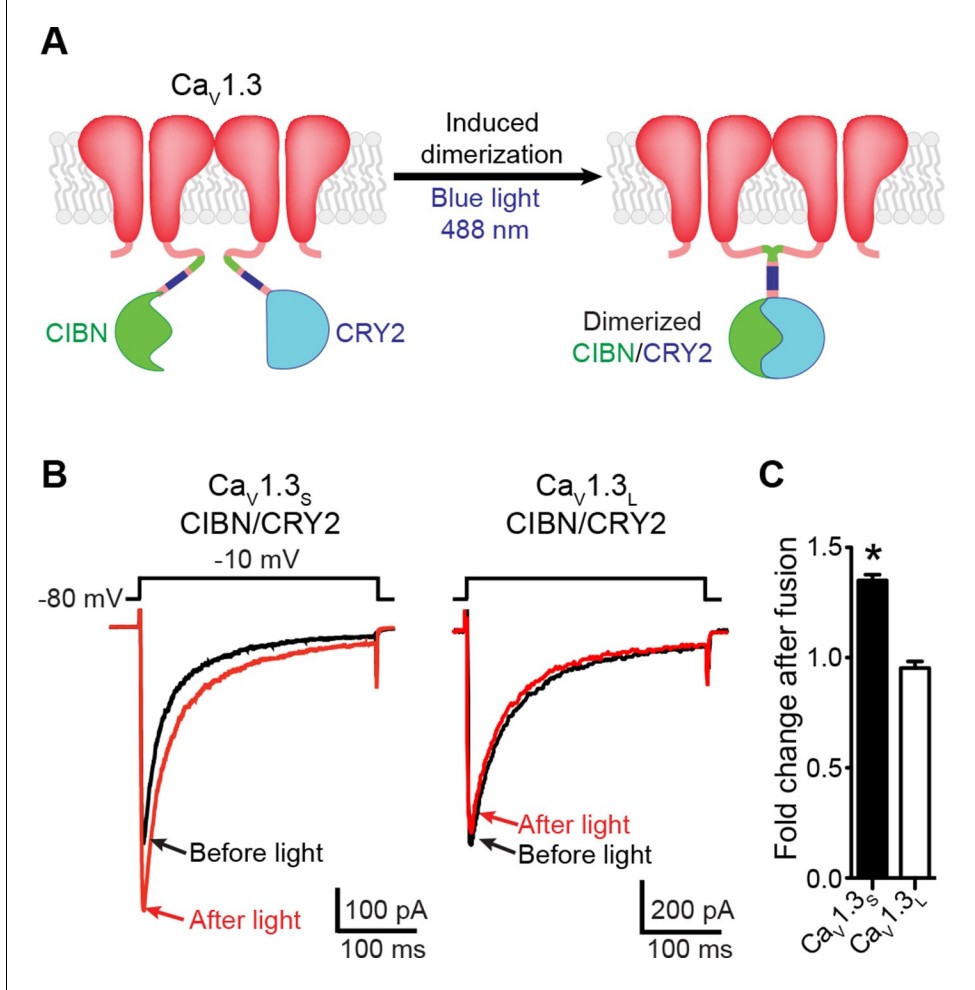

**Figure 4.** Light-induced fusion increases $I_{Ca}$ amplitude in $Ca_V1.3_S$ but not in $Ca_V1.3_L$ channels. (**A**) Schematic of the blue light-induced dimerization system (CIBN-CRY2) fused to the C-terminal of $Ca_V1.3_S$ channels. The same proteins were fused to the C-terminal of $Ca_V1.3_L$ channels (not shown in schematic). (**B**) Representative current records from tsA-201 cells expressing $Ca_V1.3_S$-CIBN/$Ca_V1.3_S$-CRY2 (*left*) or $Ca_V1.3_L$-CIBN/$Ca_V1.3_L$-CRY2 (*right*), before (*black traces*) and after (*red traces*) induction of channel coupling by excitation with a 30 s pulse of 488 nm light. (**C**) Bar plot of the averaged fold-change in $I_{Ca}$ following 488 nm excitation in cells expressing $Ca_V1.3_S$-CIBN/ $Ca_V1.3_S$-CRY2 (black) or $Ca_V1.3_L$-CIBN/$Ca_V1.3_L$-CRY2. Bars are averages of 5 cells ± SEM (*p<0.05).

change in current amplitude (0.95 ± 0.03 n = 6). These results suggest that fusing adjacent channels at the tip of their C-tail increase the probability of functional coupling between $Ca_V1.3_S$ adjoined channels but not between $Ca_V1.3_L$ channels.

## $Ca_V1.3_S$ channels couple under membrane depolarization in a $Ca^{2+}$-dependent manner

We used a second optogenetic approach that entailed fusing $Ca_V1.3_S$ and $Ca_V1.3_L$ channels with either the N- (VN) or C-terminus (VC) of the Venus fluorescent protein (*Kodama and Hu, 2010*; *Shyu et al., 2006*) (*Figure 5A*). Individual VN and VC fragments are non-fluorescent, but when they come into close proximity, they can reconstitute a full Venus protein, resulting in fluorescence. Venus reconstitution is irreversible and thus, the intensity of the fluorescence signal increases with time, proportionate to the number of $Ca_V1.3_{S/L}$-VN and $Ca_V1.3_{S/L}$-VC channels that physically associate.

We simultaneously recorded $I_{Ca}$ and obtained TIRF microscopic images from tsA-201 cells transfected with $Ca_V1.3_S$ -VN and $Ca_V1.3_S$ -VC. The first set of experiments was performed in the presence of 20 mM $Ca^{2+}$ to mimic the experimental conditions used to record $Ca^{2+}$ sparklets below.

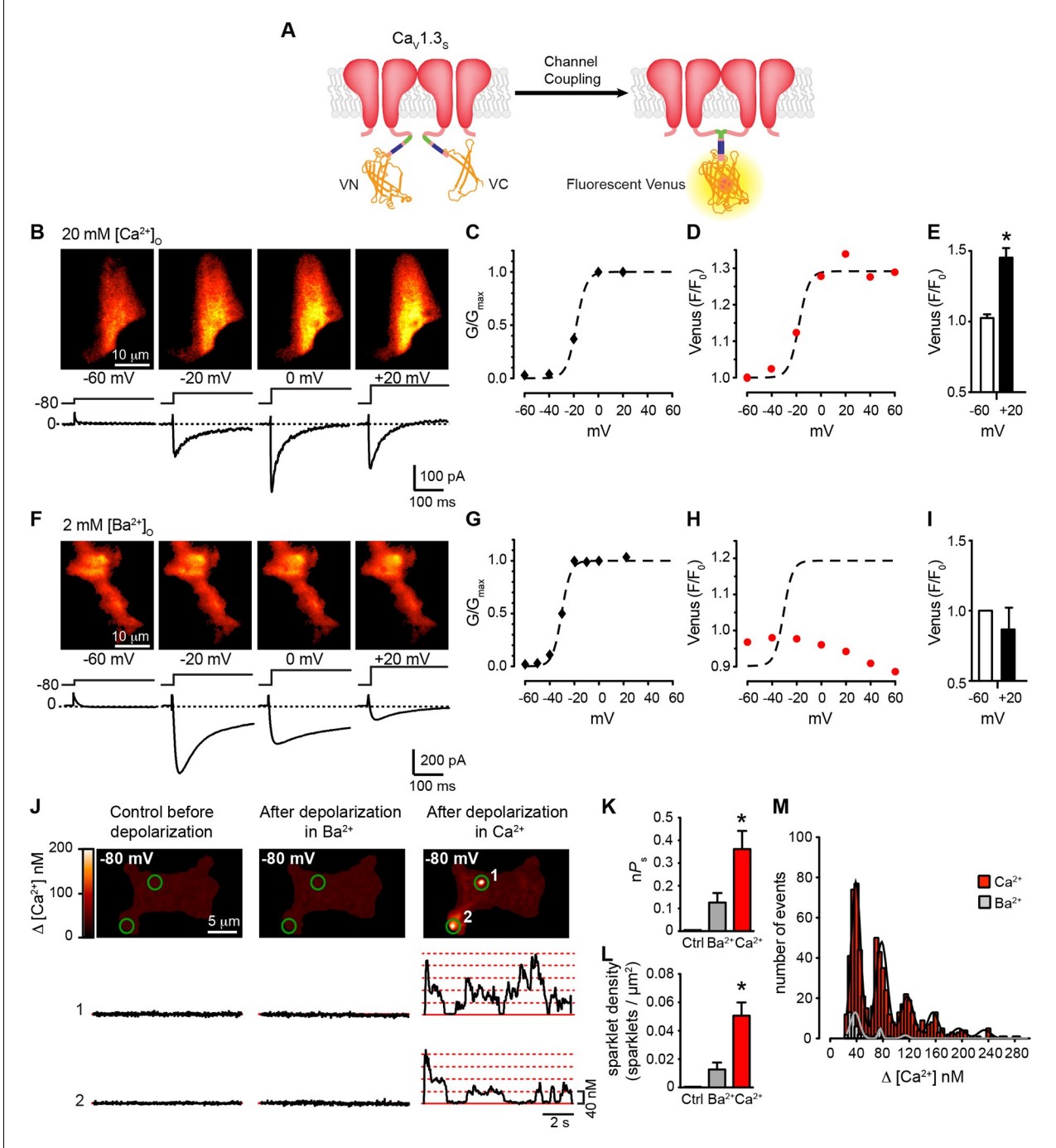

**Figure 5.** Coupling of $Ca_V1.3_S$ channels is $Ca^{2+}$-dependent and increases channel activity. (**A**) Schematic of $Ca_V1.3_S$ fused to VN and VC fragments of the Split Venus bimolecular fluorescence complementation system. (**B**) TIRF images of Venus fluorescence reconstitution in the presence of 20 mM $Ca^{2+}$ in tsA-201 cells expressing $Ca_V1.3_S$-VN and $Ca_V1.3_S$-VC (*top*). Fluorescence reconstitution was measured in response to 9-s depolarizing voltage steps from a holding potential of -80 mV to test potentials of -60 mV to +60 mV. $I_{Ca}$ currents evoked at the different voltage steps (*bottom*). (**C**) Voltage-dependence of the normalized conductance ($G/G_{max}$) of the $I_{Ca}$ shown in (**B**). Dashed curve is the fit to a Boltzmann function. (**D**) Voltage-dependence of Venus fluorescence reconstitution in the presence of 20 mM $Ca^{2+}$. The Boltzmann function calculated in (**C**) is superimposed to compare voltage-dependence. (**E**) Bar plot of averaged Venus fluorescence in the presence of 20 mM $Ca^{2+}$ at -60 mV and +20 mV. Bars are averages ± SEM (*p<0.05, n = 5 cells). (**F**) TIRF images of Venus fluorescence reconstitution in the presence of 2 mM $Ba^{2+}$ in tsA-201 cells expressing $Ca_V1.3_S$-VN and $Ca_V1.3_S$-VC (*top*). Format and protocol are as in (**B**) $I_{Ba}$ currents evoked at the different voltage steps (*bottom*). (**G**) Voltage dependence of normalized conductance ($G/G_{max}$) of the $I_{Ba}$ shown in (**F**). Dashed curve is the fit to a Boltzmann function. (**H**) Voltage dependence of Venus fluorescence reconstitution in the presence of 2 mM $Ba^{2+}$. The Boltzmann function calculated in (**G**) was superimposed to compare voltage-dependence. (**I**) Bar plot of averaged Venus

*Figure 5 continued on next page*

*Figure 5 continued*

fluorescence in the presence of 20 mM $Ca^{2+}$ at -60 mV and +20 mV. Bars are averages of 5 cells ± SEM (*p<0.05). Venus reconstitution was also tested in the presence 2 mM $Ca^{2+}$ (See *Figure 5—figure supplement 1*) (J) Top: TIRF images of $Ca_V1.3_S$ sparklets recorded at -80 mV in 20 mM $Ca^{2+}$, before depolarization (*left*), after the same depolarization protocol used in (B, F) in the presence of 2 mM $Ba^{2+}$ (*center*), and after depolarization in the presence of 20 mM $Ca^{2+}$ (*right*). Green circles indicate sparklet sites. Bottom: Traces of the time course of $[Ca^{2+}]_i$ in sites 1 and 2 under the three conditions. (K) Bar plot of the averaged $Ca_V1.3_S$ sparklet activity ($nPs$) before depolarization (black; average is ∼0), after depolarization in $Ba^{2+}$ (gray), and after depolarization in $Ca^{2+}$ (red). Bars are averages ± SEM (*p<0.05, n = 5 cells). (L) Bar plot of sparklet density. Format as in (K). (M) Event amplitude histograms of $Ca_V1.3_S$ sparklets recorded after depolarization in the presence of $Ba^{2+}$ (gray) or $Ca^{2+}$ (red). The amplitude of elementary $Ca_V1.3$ sparklets was calculated by fitting histograms to a multicomponent Gaussian function. The experiments in this figure were performed using the $Ca_V1.3_S$ channel encoded by the Addgene plasmid 26576, similar results for split Venus reconstitution and sparklet activity were observed for the plasmid 49,333 (*Figure 5—figure supplement 2*).

The following figure supplements are available for figure 5:

**Figure supplement 1.** Depolarization in the presence of physiological $Ca^{2+}$ concentrations induces coupling in $Ca_V1.3_S$.

**Figure supplement 2.** $Ca_V1.3_S$ and $Ca_V1.3_{S(G244S)}$ channels exhibit $Ca^{2+}$-dependent coupling and increased sparklet activity after depolarization.

With 20 mM $Ca^{2+}$ in the bathing solution, $I_{Ca}$ and Venus fluorescence for $Ca_V1.3_S$ channels increased in parallel in response to depolarizing pulses from a holding potential of -80 mV (*Figure 5B*). The normalized conductance and Venus fluorescence exhibited similar sigmoidal voltage dependencies (*Figures 5C–D*), which can be attributed to the irreversible nature of the Venus reconstitution, resulting in an increased number of fluorescent proteins with each successive depolarization as the open probability of $Ca_V1.3_S$ channels increases.

Substituting 2 mM $Ba^{2+}$ for 20 mM $Ca^{2+}$ in the external solution led to a robust $I_{Ba}$ during membrane depolarization, but caused no accompanying change in Venus fluorescence (*Figures 5F–I*). Importantly, Venus reconstitution was never observed in cells expressing $Ca_V1.3_L$-VN and $Ca_V1.3_L$-VC in (see Figure 8). These data indicate that $Ca_V1.3_S$ channels, but not $Ca_V1.3_L$ channels, can physically interact via their C-termini, that this association occurs in response to membrane depolarization, and that it is promoted by intracellular $Ca^{2+}$.

Because Venus reconstitution is irreversible, once $Ca_V1.3_S$-VN and $Ca_V1.3_S$-VC channels fuse they must remain adjoined. Thus, we tested the hypothesis that $Ca^{2+}$-induced fusion of $Ca_V1.3_S$-VN and $Ca_V1.3_S$-VC increases the activity of adjoined channels by recording $Ca_V1.3_S$ sparklets at a holding potential of -80 mV, in the presence of $Ba^{2+}$ or $Ca^{2+}$, before and after applying the same depolarization protocol described above. Before depolarization, $Ca_V1.3_S$ sparklet activity was very low (*Figure 5J*); after depolarization, $Ca_V1.3_S$ sparklet activity ($nP_s$; *Figure 5K*) and sparklet density (*Figure 5L*) markedly increased in the presence of $Ca^{2+}$, but not $Ba^{2+}$, without a change in the amplitude of the elementary $Ca^{2+}$ influx (*Figure 5M*).

## Functional $Ca_V1.3_S$-to-$Ca_V1.3_S$ channel coupling is mediated by physical interactions between $Ca^{2+}$-CaM and the pre-IQ domain

To investigate the mechanism underlying the $Ca^{2+}$-dependency of $Ca_V1.3_S$ channel coupling further, we used the Venus system in conjunction with a mutagenesis approach, focusing on CaM, which is required for CDF and has been shown to bind $Ca^{2+}$ and associate with the C-terminus of L-type $Ca^{2+}$ channels (*Zuhlke et al., 1999*). tsA-201 cells were transfected with $Ca_V1.3_S$-VN/$Ca_V1.3_S$-VC and divided into three groups. Cells in the first group (controls) were dialyzed with a standard $Cs^+$-based intracellular solution. Cells in the second group were dialyzed with a CaM-inhibitory peptide corresponding to a 15-aa fragment of the wild-type CaM-binding domain of myosin light chain kinase (MLCKp; 1 µM), which binds to CaM with high affinity (apparent dissociation constant, ∼6 pM) in the presence of $Ca^{2+}$ (*Torok and Trentham, 1994*) and has been used by others as a competitive inhibitor of CaM (*Ciampa et al., 2011*; *Mercado et al., 2010*; *Piper and Large, 2004*). The third group consisted of cells co-expressing a dominant-negative mutant form of CaM ($CaM_{1234}$) that does not bind $Ca^{2+}$ through its N- or C-terminal lobes. Dialysis of MLCKp or co-expression of $CaM_{1234}$ prevented $Ca_V1.3_S$-VN and $Ca_V1.3_S$-VC fusion upon membrane depolarization (*Figure 6A–D*). Although MLCKp and $CaM_{1234}$ were equally effective in preventing Venus reconstitution, they had differential effects on $I_{Ca}$ inactivation (*Figure 6E*). Whereas the fraction of peak $I_{Ca}$ remaining at

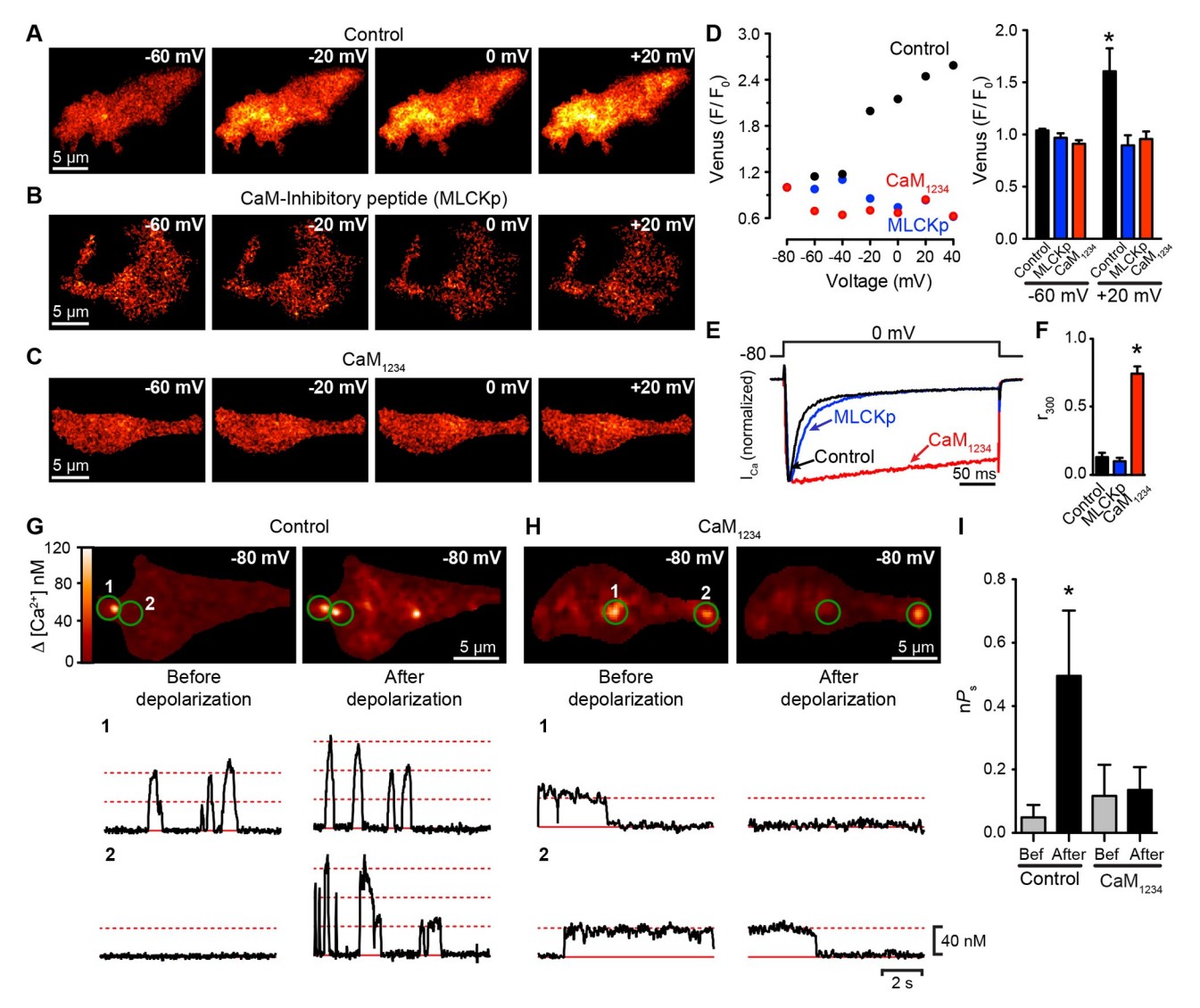

**Figure 6.** $Ca_V1.3_S$ coupling requires $Ca^{2+}$-CaM. (**A–C**) TIRF images of Venus fluorescence reconstitution in the presence of 20 mM $Ca^{2+}$ in tsA-201 cells expressing (**A**) $Ca_V1.3_S$-VN and $Ca_V1.3_S$-VC, (**B**) $Ca_V1.3_S$-VN and $Ca_V1.3_S$-VC and dialyzed with the MLCK peptide (MLCKp), (**C**) $Ca_V1.3_S$-VN, $Ca_V1.3_S$-VC and $CaM_{1234}$. Fluorescence reconstitution was measured in response to depolarizing voltage steps from a holding potential of -80 mV to test potentials of -60 mV to +60 mV. (**D**) Voltage-dependence of Venus fluorescence reconstitution in the presence of 20 mM $Ca^{2+}$ for control (black), MLCKp (blue), and $CaM_{1234}$ (red) cells shown in (**A–C**) (*left*). Bar plot of averaged Venus fluorescence in the presence of 20 mM $Ca^{2+}$ at -60 mV and +20 mV (*right*). Bars are averages ± SEM (*p<0.05, n = 5 cells). (**E**) Normalized $I_{Ca}$ currents evoked by a 300-ms depolarizing pulse from a holding potential of -80 mV to a test potential of 0 mV in control (black), MLCKp (blue), and $CaM_{1234}$ (red) cells. Currents analyzed for these experiments were in a range between 0.3 and 1.2 nA (**F**) Bar plot of the mean fraction of $r_{300}$ at 0 mV. Bars are averages ± SEM (*p<0.05, n = 5 cells). (**G**) Top: TIRF images of $Ca_V1.3_S$ sparklets in tsA-201 cells expressing $Ca_V1.3_S$-VN and $Ca_V1.3_S$-VC (Control). Sparklets were recorded at -80 mV in 20 mM $Ca^{2+}$ before depolarization (left) and after the same depolarization protocol used in (**A–C**) (right). Green circles indicate sparklet sites. Bottom: Traces of the time course of $[Ca^{2+}]_i$ in the corresponding sparklet sites 1 and 2. (**H**) TIRF images and time course of $[Ca^{2+}]_i$ of $Ca_V1.3_S$ sparklets in tsA-201 cells expressing $Ca_V1.3_S$-VN/$Ca_V1.3_S$-VC and $CaM_{1234}$. Format and protocol are as in (**G**). (**I**) Bar plot of the averaged $Ca_V1.3_S$ sparklet activity ($nP_s$) before (gray) and after (black) depolarization. Bars are averages 5 cells ± SEM (*p<0.05).

300 ms ($r_{300}$) in MLCKp-dialyzed cells (0.11 ± 0.03, n = 5) was similar to that of controls (0.14 ± 0.03, n = 5), the rate of inactivation of $I_{Ca}$ was slower in cells expressing $CaM_{1234}$, as reflected in a much higher $r_{300}$ value (0.75 ± 0.05, n = 5) (*Figure 6F*). Our interpretation of these findings is that the CaM molecules involved in CDI are distinct from those involved in functional coupling of $Ca_V1.3_S$ channels. The results suggest that CaM molecules that mediate coupling could be both attached to the

channels or recruited to the C-terminus during depolarization. This could explain why they are accessible to MLCKp blockade, unlike the CaM molecules that mediate CDI, which are tethered to the IQ domain of the channels (*Pitt et al., 2001*).

To extend this analysis, we recorded $Ca_V1.3_S$ sparklets in control and $CaM_{1234}$ cells before and after depolarization to +20 mV (*Figure 6G and H*). Although multi-quantal $Ca_V1.3_S$ sparklets were observed in control cells at rest (e.g., *Figure 6G*, trace 2 left), sparklets in $CaM_{1234}$ cells prior to depolarization were long, single-quantal events (*Figure 6H*, traces 1 and 2, left). These long $Ca_V1.3_S$ sparklets were likely due to decreased CDI of $Ca_V1.3_S$ channels in cells expressing $CaM_{1234}$ (*Yang et al., 2014*). Importantly, the overall $Ca_V1.3_S$ sparklet density was lower in cells expressing $CaM_{1234}$ than WT, suggesting that Apo-CaM does not increase $Ca_V1.3_S$ channel activity.

The coupling coefficient for $Ca_V1.3_S$ sparklet sites in WT cells was $0.07 \pm 06$, whereas that in $CaM_{1234}$ cells was $0.02 \pm 0.02$ (n = 5). Membrane depolarization increased the coupling coefficient of $Ca_V1.3_S$ channels within multi-quantal sparklet sites in control cells ($0.18 \pm 0.03$, n = 5), but not in $CaM_{1234}$ cells ($0.08 \pm 0.05$, n = 5). The opposing effects of $CaM_{1234}$ on $Ca_V1.3_S$ sparklets — longer, but decoupled — resulted in $Ca_V1.3_S$ sparklet activity before depolarization that was similar in control ($nP_S = 0.07 \pm 0.05$; p>0.05) and $CaM_{1234}$ ($nP_S = 0.12 \pm 0.09$) cells (*Figure 6I*). However, depolarization increased $Ca_V1.3$ sparklet activity nearly 7-fold in control cells, but had no effect on sparklet activity in $CaM_{1234}$ cells. Taken together with the $I_{Ca}$ and Venus reconstitution results described above, these data strongly suggest that $Ca^{2+}$ binding to CaM is required for physical and functional $Ca_V1.3_S$-to-$Ca_V1.3_S$ channel coupling.

Finally, to establish the molecular mechanism by which CaM might mediate these effects, we mutated different CaM-binding domains of the $Ca_V1.3_S$. L-type channels have two binding sites for CaM in their C-terminus: the IQ domain (aa K1601-Q1621) and the pre-IQ domain (aa T1545-Q1587) (*Fallon et al., 2009*). These sites have different affinities for CaM; whereas CaM is "pre-associated" and binds tightly to the IQ domain, the association of CaM with the pre-IQ domain is seemingly weaker and likely transient. To determine which of these sites is necessary for CaM-mediated $Ca_V1.3_S$ coupling, we generated two mutants of the $Ca_V1.3_S$-VN and $Ca_V1.3_S$-VC channels. The first contained a single point mutation I1608E ($Ca_V1.3$-I1608E) that disrupts CaM binding to the IQ domain (*Zuhlke et al., 1999*), and the second contained a triple mutation (L1569A, V1572A, and W1577E; $Ca_V1.3_S$-AAE) that prevents CaM binding to the pre-IQ domain and anti-parallel coiled-coil arrangement of the pre-IQ domains (*Fallon et al., 2009*) (*Figure 7A*). $Ca_V1.3_S$-I1608E channels showed a slower rate of inactivation than control and $Ca_V1.3_S$-AAE channels (*Figure 7B*), consistent with the lack of CDI. Interestingly, $Ca_V1.3_S$-I1608E-VN/VC, but not $Ca_V1.3_S$-AAE VN/VC, which were capable of reconstituting Venus during membrane depolarization (*Figure 7C–G*). These data suggest that binding of CaM to the pre-IQ domain is critically involved in $Ca_V1.3_S$ channel coupling during membrane depolarization and further supports our previous assertion that the CaM pool involved in CDI is distinct from that involved in channel coupling.

## Deletion of DCRD is not sufficient to allow coupling in $Ca_V1.3_L$ channels

We investigated one of the possible molecular mechanisms that prevent functional coupling in $Ca_V1.3_L$ channels. *Singh et al (2008)* showed that a $Ca_V1.3_L$ mutant lacking the last 116 amino acids of the C-terminus ($Ca_V 1.3_L\Delta116$) had voltage-dependencies and kinetics similar to those of $Ca_V1.3_S$ channels. The $Ca_V1.3_L\Delta116$ channels lack the distal regulatory domain (DCRD) that binds the proximal regulatory domain (PCRD) located downstream to the pre-IQ and IQ domains of the channel (*Figure 8A*). Interaction of these two regulatory domains interferes with CaM binding to the IQ domain and results in a reduction in CDI (*Bock et al., 2011*; *Singh et al., 2008*). If the DCRD interferes also with the binding of CaM to the pre-IQ domain, which we propose is important for channel-to-channel coupling, we would expect that removing the DCRD would allow coupling between $Ca_V1.3_L$ channels. Thus, we investigated whether or not $Ca_V1.3_L\Delta116$ channels are capable of undergoing $Ca^{2+}$-driven physical interactions. For these experiments, we created $Ca_V1.3_L\Delta116$ channels fused to the split Venus system. As expected, CDI of $I_{Ca}$ was faster in cells expressing the $Ca_V1.3_L\Delta116$ channels compared to the full length $Ca_V1.3_L$ channels (p<0.05; *Figure 8B and C*).

We found that, with 2 mM $Ca^{2+}$ in the bathing solution, cells expressing $Ca_V1.3_L\Delta116$ channels failed to reconstitute Venus fluorescence, similar to what we observed for the full length $Ca_V1.3_L$ channel (*Figure 8D–G*). Even increasing the extracellular $Ca^{2+}$ concentration to 20 mM was not enough to induce coupling in the $Ca_V1.3_L\Delta116$ channels (*Figure 8—figure supplement 1*, n = 5,

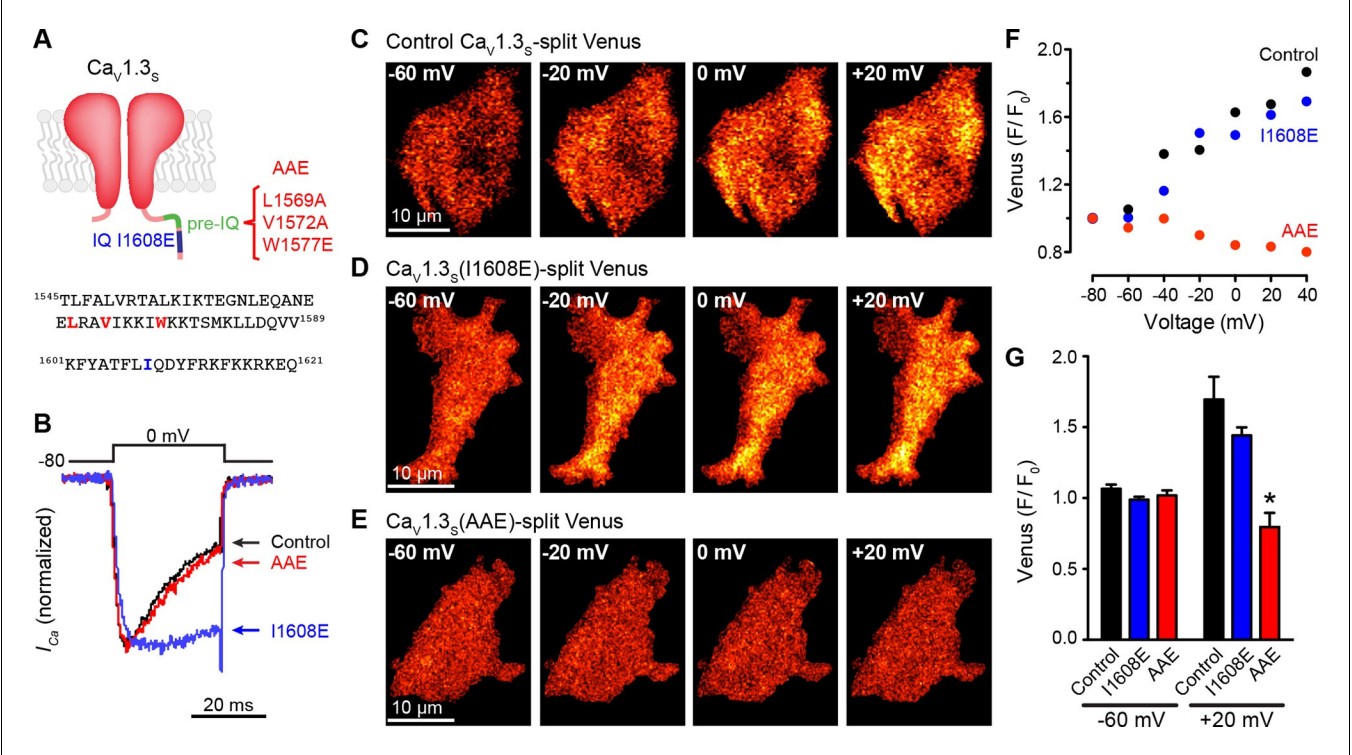

**Figure 7.** The pre-IQ domain is required for $Ca^{2+}$-CaM-mediated $Ca_V1.3_S$ coupling. (**A**) Schematic of $Ca_V1.3_S$ mutations introduced to disrupt CaM binding to the IQ (I1608E) or the pre-IQ (AAE) domain; the position of the mutated amino acid is shown in the sequence below. (**B**) Normalized $I_{Ca}$ currents evoked by a 30-ms depolarizing pulse from a holding potential of -80 mV to a test potential of 0 mV in tsA-201 cells expressing $Ca_V1.3_S$ (Control, black), $Ca_V1.3_S$(I1608E) (blue), or $Ca_V1.3_S$(AAE) (red). Currents analyzed for these experiments were in a range between 100 and 600 pA (**C–E**) TIRF images of Venus fluorescence reconstitution in the presence of 20 mM $Ca^{2+}$ in tsA-201 cells expressing (**C**) $Ca_V1.3_S$-VN and $Ca_V1.3_S$-VC, (**D**) $Ca_V1.3_S$(I1608E)-VN and $Ca_V1.3_S$(I1608E)-VC, or (**E**) $Ca_V1.3_S$(AAE)-VN and $Ca_V1.3_S$(AAE)-VC. Fluorescence reconstitution was measured in response to depolarizing voltage steps from a holding potential of -80 mV to test potentials of -60 mV to +60 mV. (**F**) Voltage-dependence of Venus fluorescence reconstitution in the presence of 20 mM $Ca^{2+}$ for control (black), I1608E mutant (blue), and AAE mutant (red) from the cells shown in (**C–E**). (**G**) Bar plot of averaged Venus fluorescence in the presence of 20 mM $Ca^{2+}$ at -60 mV and +20 mV. Bars are averages of 5 cells ± SEM (*p<0.05).

p=0.245 -60 mV vs 20 mV). This result suggests that deletion of DCRD is not sufficient to allow coupling of $Ca_V1.3_L$ channels. As $Ca_V1.3_L\Delta116$ channels still have a C-terminus (396 aa) that is considerably longer than that of $Ca_V1.3_S$ channels, it is possible that another domain inside this region might be responsible for occluding the binding of CaM to the pre-IQ domain, and preventing $Ca_V1.3_L\Delta116$ channel coupling. Differential folding between the short and long C-terminal could be another explanation for the ability of $Ca_V1.3_S$ channel-to-channel coupling during depolarization-induced $Ca^{2+}$ entry.

## Coupling of $Ca_V1.3_S$ channels increases the firing rate of hippocampal neurons

We extended our investigation of the functional consequences of $Ca_V1.3_S$ channel coupling to cultured rat hippocampal neurons. We began by recording spontaneous $Ca_V1.3$ sparklets from these cells. As in tsA-201 cells expressing $Ca_V1.3_S$ channels, $Ca^{2+}$ sparklets in neurons were restricted to specific sites and had multi-quantal amplitudes resulting from the simultaneous opening and closing of multiple channels. The quantal unit of $Ca^{2+}$ influx was about 40 nM. The coupling coefficient (κ) of sparklet sites ranged from 0.14 to 0.33. The average κ value was 0.23 ± 0.04 (n = 6). Importantly, sparklet site activity was decreased by application of 300 nM of the dihydropyridine antagonist isradipine and completely eliminated when the concentration of the drug was increased to 10 μM (*Figure 9A and B*). This is consistent with the hypothesis that sparklets in hippocampal neurons were produced by L-type calcium channels. Both $Ca_V1.2$ and $Ca_V1.3$ channels are expressed in

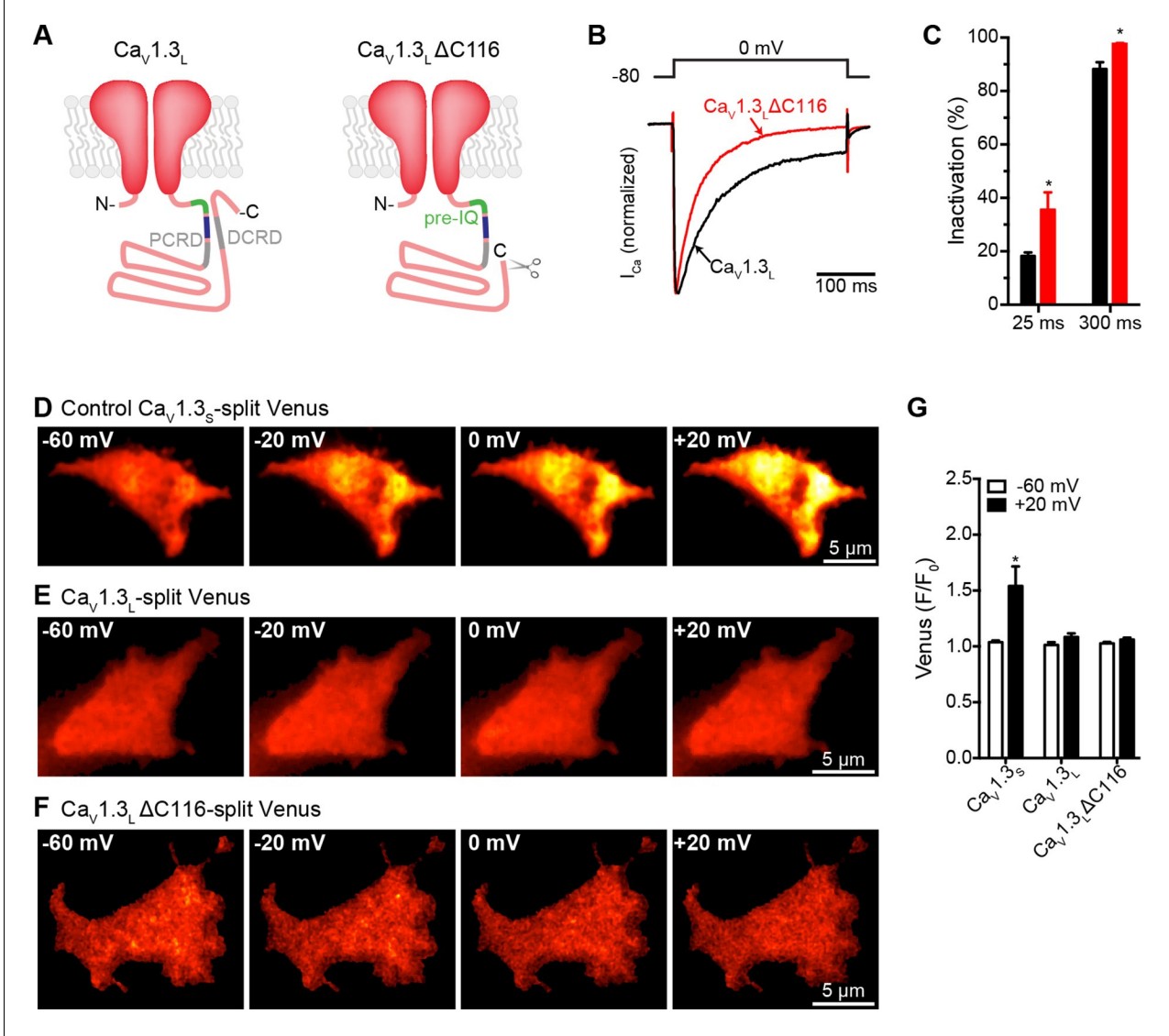

**Figure 8.** Distal auto-regulatory domain (DCRD) is not responsible for the lack of coupling of $Ca_V1.3_L$ channels. (A) Schematic of $Ca_V1.3_L$ channel splice variant (*left*), depicting the domains important for $Ca^{2+}$-mediated regulation: pre-IQ (green), IQ (blue), proximal and distal C-terminal regulatory domains (PCRD, DCRD, gray). Schematic of the $Ca_V1.3_L\Delta C116$ channel where the last 116 aa in the C-terminal were removed (*right*). (B) Representative currents of $Ca_V1.3_L$ (black) and $Ca_V1.3_L$ $\Delta C116$ channels (red) expressed in tsA-201 cells. Currents were evoked by a 300-ms depolarization from holding potential of -80 mV to a test potential of 0 mV, with 2 mM $Ca^{2+}$ as the charge carrier. Currents analyzed for these experiments were in a range between 0.3 and 1 nA (C) Bar plot of the% inactivation after 25 or 300 ms at 0 mV. Bars are averages of 5 cells ± SEM (*p < 0.001) (D–F) TIRF images of Venus fluorescence reconstitution in the presence of 2 mM $Ca^{2+}$ in tsA-201 cells expressing $Ca_V1.3_S$-VN and $Ca_V1.3_S$-VC (D) $Ca_V1.3_L$-VN and $Ca_V1.3_L$-VC (E) or $Ca_V1.3_L$ $\Delta C116$-VN and $Ca_V1.3_L$ $\Delta C116$-VC (E). Fluorescence reconstitution was measured in response to depolarizing voltage steps from a holding potential of -80 mV to test potentials of -60 mV to +60 mV. (G) Bar plot of averaged Venus fluorescence at -60 mV and +20 mV for each of the aforementioned construct pairs. Bars are averages of 5 cells ± SEM (*p<0.05). Data for $Ca_V1.3_L$ $\Delta C116$-VN and $Ca_V1.3_L$ $\Delta C116$-VC Venus reconstitution with 20 mM $Ca^{2+}$ is presented in *Figure 8—figure supplement 1*.

The following figure supplement is available for figure 8:

**Figure supplement 1.** High Ca2+ concentration is not enough to induce coupling in $Ca_V1.3_L\Delta C116$ channels.

hippocampal neurons (*Hell et al., 1993*) and although it is impossible to distinguish between these two L-type $Ca^{2+}$ channels using either electrophysiological or pharmacological methods, it has been shown that $Ca_V1.3$ channels have a reduced sensitivity to dihydropyridines compared to $Ca_V1.2$ channels (*Lipscombe et al., 2004*; *Xu and Lipscombe, 2001*). A previous study by Koschak et al

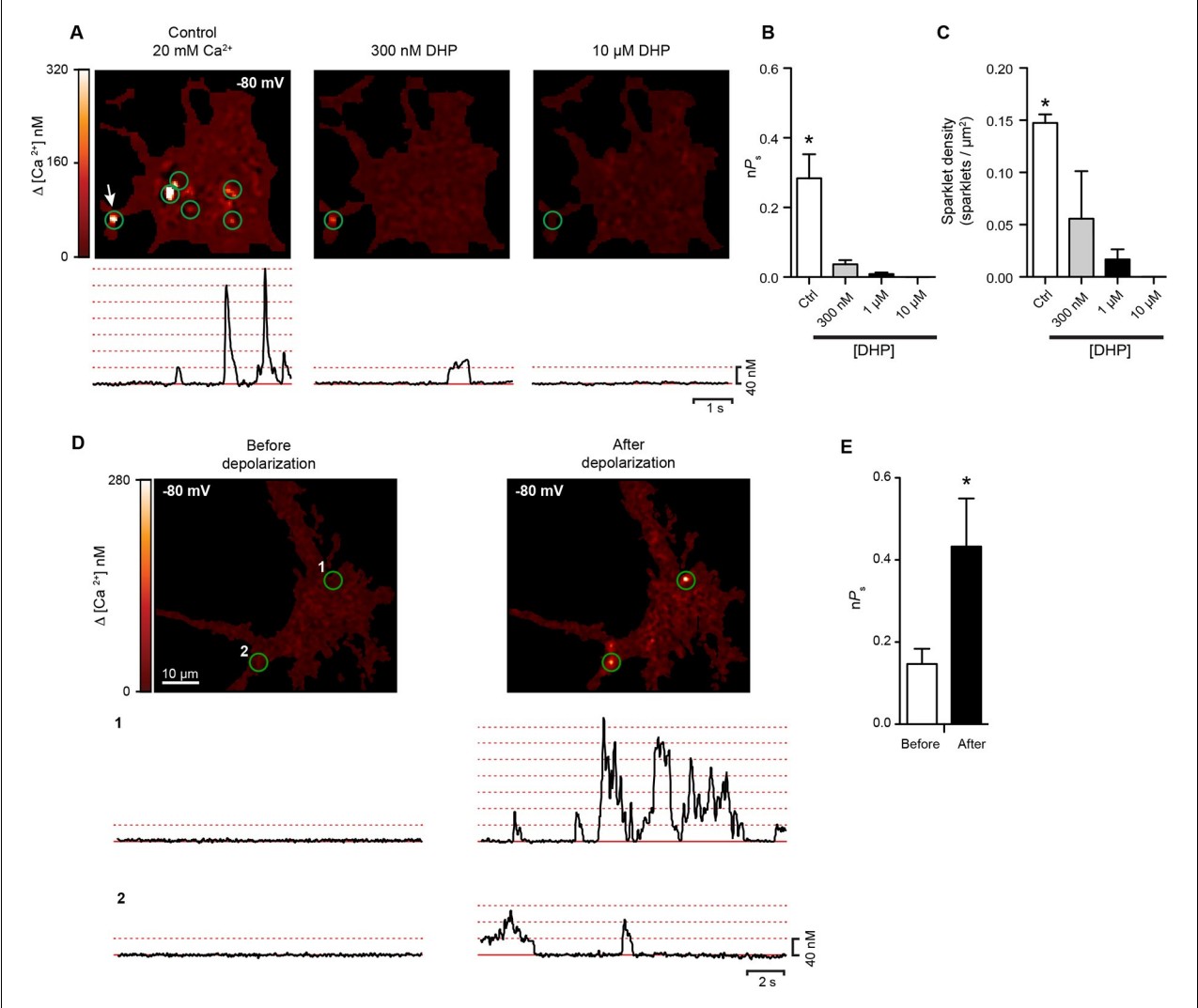

**Figure 9.** Hippocampal neurons exhibit dihydropyridine-sensitive spontaneous persistent sparklet activity that is increased after depolarization. (**A**) *Top:* TIRF images of $Ca^{2+}$ sparklets recorded at -80 mV in cultured hippocampal neurons (4 div) under control conditions (20 mM $[Ca^{2+}]_o$; *left*), and after exposure to low (1 μM; *middle*) and high (10 μM; *right*) concentrations of dihydropyridine (DHP). Green circles indicate sparklet sites. Bottom: Traces of the time course of $[Ca^{2+}]_i$ at the site indicated by the white arrow for each condition are shown below the relevant image. Dotted red lines show the amplitudes of 1 to 7 quantal levels. (**B**) Bar chart showing the mean $Ca^{2+}$ sparklet activity (*nPs*) in control conditions and after exposure to 300 nM, 1 μM or 10 μM concentrations of DHP. (**C**) Bar chart showing sparklet density for each condition described in (**B**) (*p<0.05, n = 6 cells). (**D**) *Top*: $Ca^{2+}$ sparklets recorded at -80 mV in cultured hippocampal neurons (4 div), before depolarization (*left*) and after depolarization (*right*). Green circles indicate sparklet sites. Bottom: Traces of the time course of $[Ca^{2+}]_i$ in sites 1 and 2 before and after depolarization. (**E**) Bar plot of the averaged $Ca^{2+}$ sparklet activity (*nPs*) before (white) and after depolarization (black). Bars are averages of 4 cells ± SEM (*p<0.05).

found that 100% of $Ca_V1.2$ channels but only ~60% of $Ca_V1.3$ channels are inhibited by 300 nM isradipine (*Koschak et al., 2001*), given this, it is reasonable to assume that any sparklet remaining after application of 300 nM isradipine is more likely to be generated by $Ca_V1.3$ channels than $Ca_V1.2$. In addition, near by 25% of the L-type current in hippocampal neurons is carried by $Ca_V1.3$ channels (*Moosmang et al., 2005*), this proportion is in agreement with the remaining sparklet site density we observed after the treatment with 300 nM isradipine (*Figure 9C*).

As was the case for tsA-201 cells expressing $Ca_V1.3_S$ channels, we found that conditioning membrane depolarization to 0 mV increased sparklet activity nearly 3-fold in the hippocampal neurons and induced persistent sparklet activity (*Figure 9D and E*). The coupling coefficient increased from 0.15 ± 0.04 to 0.43 ± 0.12 after membrane depolarization (n = 8). These results support the

hypothesis that L-type channels undergo cooperative gating, generating persistent $Ca^{2+}$ influx in hippocampal neurons. The persistent L-type channels sparklet activity evoked by membrane depolarization in the presence of $Ca^{2+}$ bears a striking resemblance to the persistent cationic currents observed in several types of neurons (*Fransen et al., 2006*; *Major and Tank, 2004*; *Moritz et al., 2007*; *Powers and Binder, 2001*).

We found that the somatic and dendritic membranes of neurons expressing both $Ca_V1.3_S$-VN and $Ca_V1.3_S$-VC channels displayed prominent Venus fluorescence (*Figure 10A*), indicating fusion of $Ca_V1.3_S$-VN and $Ca_V1.3_S$-VC channel pairs and functional Venus reconstitution. Since Venus reconstitution is irreversible and neurons have spontaneous electrical activity, this observation suggests that the C-termini of $Ca_V1.3_S$ channels make contact during normal neuronal firing. Although spontaneous self-assembly between Venus subunits might conceivably drive an interaction that would not otherwise occur, the improved Venus system used here has a mutation (I152L) that minimizes nonspecific interactions (*Kodama and Hu, 2010*).

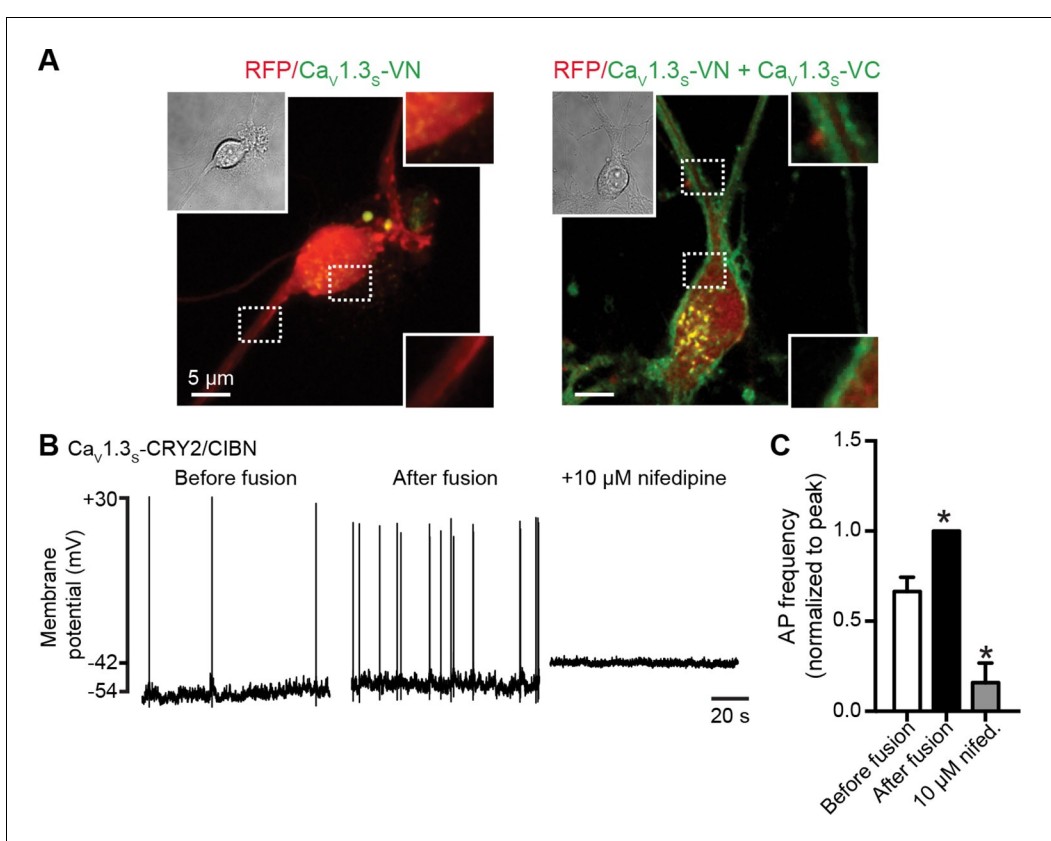

**Figure 10.** $Ca_V1.3_S$ coupling increases the firing rate of hippocampal neurons. (**A**) Confocal images of two representative cultured hippocampal neurons expressing tRFP as a transfection marker (red) and $Ca_V1.3_S$-VC (*left*, negative control) or $Ca_V1.3_S$-VC/$Ca_V1.3_S$-VN (*right*). Fluorescence of the spontaneously reconstituted Venus is shown in green. The insets show expanded views of the soma and dendritic regions marked by the dashed boxes. Overexpression of these channels does not change the cluster size observed with super-resolution microscopy (see also *Figure 10—figure supplemental 1*). (**B**) Representative traces of spontaneous action potentials recorded from neurons expressing $Ca_V1.3_S$-CRY2 and $Ca_V1.3_S$-CIBN before (*left*) and after (*middle*) the induction of fusion with 488 nm light and after subsequent treatment with 10 μM nifedipine (*right*). (**C**) Bar plot showing the AP frequency (normalized to the peak frequency) for each condition. Bars are averages of 4 cells ± SEM (*p<0.01).

The following figure supplement is available for figure 10:

**Figure supplement 1.** $Ca_V1.3_S$ overexpression in hippocampal neurons increased the number of $Ca_V1.3_S$ channels, but not the cluster size.

We analyzed super resolution images of hippocampal neurons transfected with $Ca_V1.3_S$-VN and $Ca_V1.3_S$-VC and found that expression of these channels increased the number of $Ca_V1.3$ channel clusters in the cells, but not the size of the clusters (*Figure 10—figure supplemental 1*). These results indicate that the spontaneous fusion of $Ca_V1.3_S$ channels was not a consequence of their being over-expressed in the plasma membranes of the hippocampal neurons.

A testable prediction of these observations is that $Ca_V1.3_S$ channel fusion—forced coupling—should augment the inward $I_{Ca}$ and consequently increase neural excitability and firing rate. *Figure 10B* shows the spontaneous action potentials recorded from a neuron transfected with $Ca_V1.3_S$-CIBN and $Ca_V1.3_S$-CRY2, before (right trace) and after (center trace) exposure to 488 nm light. Forced coupling of $Ca_V1.3_S$-CIBN and $Ca_V1.3_S$-CRY2 channel pairs increased the average firing rate by about 40% (p<0.01, *Figure 10C*). 488 nm illumination produced no effect on firing rate in control hippocampal neurons transfected only with $Ca_V1.3_S$-CIBN channels (0.90 ± 0.04, N = 4). Adding the L-type calcium channel blocker nifedipine 10 μM to the bathing solution abolished all AP activity after light-induced fusion of $Ca_V1.3_S$-CIBN and $Ca_V1.3_S$-CRY2 (*Figure 10B*, left trace and *10C*). Together, these results highlight the crucial role that $Ca_V1.3_S$ channels play in regulating the action potential firing in hippocampal neurons.

## Discussion

We have found that $Ca_V1.3$ channels in the plasma membrane of hippocampal neurons are arranged in clusters containing multiple (~8) channels. This clustering has important physiological consequences for the short-splice variant of the channel ($Ca_V1.3_S$), enabling proximal channels to engage in cooperative gating, generating more persistent and greater $Ca^{2+}$ influx. Functional coupling of $Ca_V1.3_S$ channels is promoted by intracellular $Ca^{2+}$ and involves physical interactions via channel C-termini mediated by physical interactions between $Ca^{2+}$-CaM and the pre-IQ domain. We propose that $Ca^{2+}$-driven $Ca_V1.3_S$ channel coupling constitutes a novel feed-forward mechanism for the activation of these channels during membrane depolarization and provides an apparatus for $Ca^{2+}$-dependent facilitation of inward L-type $Ca^{2+}$ currents. Further, these findings challenge a fundamental tenet of the classic Hodgkin-Huxley model of ion channel gating: that voltage-gated channels open and close independently.

Our data suggest that $Ca_V1.3_S$ channel coupling is critically dependent on intracellular $Ca^{2+}$ and CaM. During membrane depolarization, $Ca_V1.3_S$ channels open. $Ca^{2+}$ flowing through these channels creates a local increase in $[Ca^{2+}]_i$ — a $Ca_V1.3$ sparklet — near the mouth and C-terminus of the channel where the $Ca^{2+}$-binding protein CaM resides. Upon binding $Ca^{2+}$, CaM associates with the pre-IQ domain of the channels, enabling the formation of $Ca_V1.3_S$-$Ca_V1.3_S$ 'couplets'. The observation of pre-IQ dimers of the structurally similar $Ca_V1.2$ channel undergoing coiled-coil interactions in vitro (*Fallon et al., 2009*) and functionally couples to neighboring channels (*Dixon et al., 2015*) gives credence to this model. An interesting question suggested by this model is which CaM pool is involved in $Ca_V1.3_S$ channel coupling? The observation that $CaM_{1234}$ and MLCKp prevent $Ca_V1.3_S$ coupling would suggest that soluble as well as apo-CaM pre-associated with the channel could be involved in inducing physical channel-to-channel interactions. Because physically coupled $Ca_V1.3_S$ channels exhibit higher open probabilities, the overall activity of $Ca_V1.3_S$ channels within a cluster would then depend on the number of channels forming dimers or the probability of the formation of higher order oligomers. A schematic summary of our model is presented in *Figure 11*.

It is important to note that Minor and colleagues reported the formation of symmetric dimers of pre-IQ domains bridged by two CaM molecules (*Kim et al., 2010*). However, they failed to obtain any clustering when the channels were expressed in *Xenopus* oocytes. The cause for this apparent lack of clustering of $Ca_V1.2$ channels in the frog egg are unclear, but suggest that clustering and functional coupling may be features of $Ca_V1$ channels expressed only in mammalian cells, as we have shown here for $Ca_V1.3_S$ channels.

Our results suggest that while close proximity is necessary, it is certainly not sufficient to allow channel interactions. Super-resolution imaging shows that $Ca_V1.3_L$ and $Ca_V1.3_S$ channels form clusters of similar size along the surface membrane, but $Ca_V1.3_L$ channels, unlike $Ca_V1.3_S$ channels, do not undergo physical and functional coupling. Our experimental results with $Ca_V1.3_L\Delta116$ channels open the question as to whether there is another regulatory domain inside the long C-terminus, capable of blocking the interaction of CaM with the pre-IQ domain. These results also suggest that

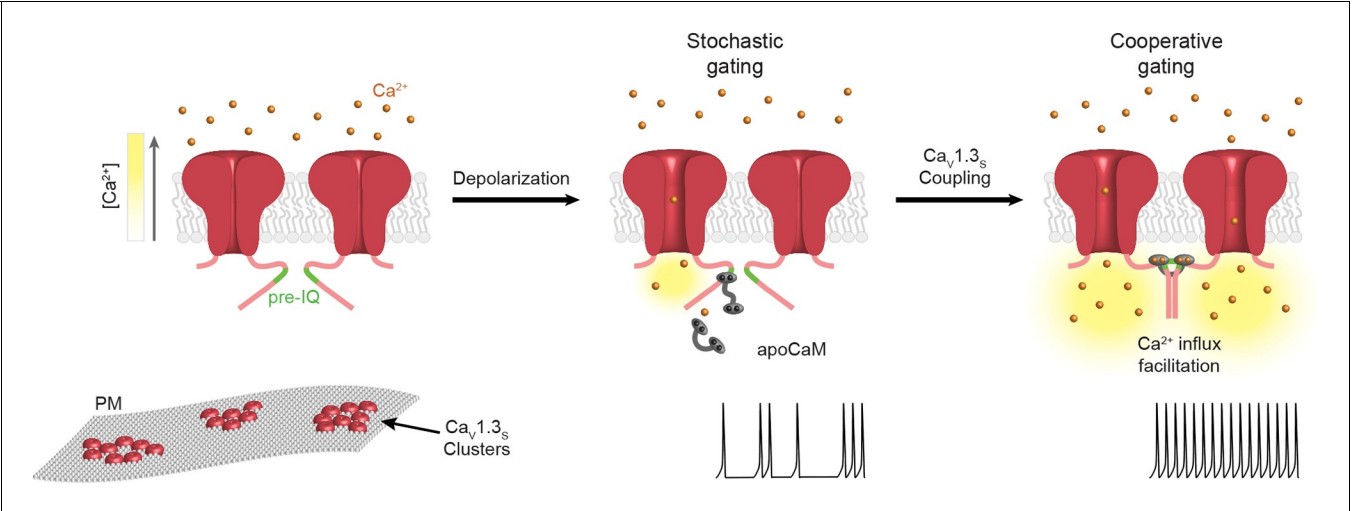

**Figure 11.** Proposed mechanism for Ca²⁺-induced functional Ca$_V$1.3$_S$ coupling in hippocampal neurons. Ca$_V$1.3$_S$ channels are organized in clusters in the plasma membrane (SM) of hippocampal neurons. At hyperpolarized potentials (e.g., -80 mV), where [Ca²⁺]$_i$ and P$_o$ of Ca$_V$1.3$_S$ channels is very low, the number of coupled channels is very low. Membrane depolarization increases the probability of stochastic (i.e., uncoupled) openings of Ca$_V$1.3$_S$ channels. Ca²⁺ flow through these channels creates a local increase in [Ca²⁺]$_i$ (yellow gradient). This Ca²⁺ binds to apoCaM, which can be tethered to the pre-IQ domain or soluble in the cytoplasm. Once CaM is activated it promotes channel-channel interaction at the pre-IQ domain. Upon association, the activity of adjoined channels increases, entering into a cooperative gating mode that facilitates Ca²⁺ influx and underlies the 'depolarizing drive' that sustain repetitive firing.

the mechanism allowing neighboring Ca$_V$1.3$_S$ channels to interact seems to be structural rather than organizational.

Although Ca$_V$1.3 channels are generally classified as high-voltage activated channels, their lower activation range allows them to generate subthreshold depolarizations that support repetitive firing (*Olson et al., 2005*). This is in particular true for the short variants of Ca$_V$1.3, which are more voltage-sensitive than Ca$_V$1.3$_L$ channels (*Bock et al., 2011*; *Tan et al., 2011*). Our results reveal a new and striking characteristic of Ca$_V$1.3$_S$ channels: they can coordinate their openings through a physical interaction to facilitate Ca²⁺ entry at low membrane potentials. In addition, the fact that Ca$_V$1.3 channels are clustered in dendritic spines in these neurons (*Gao et al., 2006*; *Jenkins et al., 2010*) raises the possibility that their coordinated gating and boosting of Ca²⁺ entry may have a significant effect on synaptic plasticity.

Subthreshold activation of Ca$_V$1.3 channels is also a key component of pacemaking and oscillatory behavior in neurons in the brain, such as dopaminergic neurons in the substantia nigra, the principal neurons affected in Parkinson's disease (*Guzman et al., 2009*; *Puopolo et al., 2007*). In this context, it is tempting to speculate that alterations in the cooperative gating of Ca$_V$1.3$_S$ might contribute to the Ca²⁺ excitotoxicity observed in multiple pathological conditions, including Parkinson's neurodegeneration. Future studies should examine the extent to which the cooperative gating of Ca$_V$1.3$_S$ channels can affect the Ca²⁺ load in neurons.

In summary, our data indicate that Ca²⁺-driven physical interactions among clustered Ca$_V$1.3$_S$ channels lead to cooperative gating of these channels and the enhanced Ca²⁺ influx that underlies the self-sustained firing of hippocampal neurons. It is anticipated that future studies may reveal that cooperative Ca$_V$1.3$_S$ channel gating plays an important role in pathological conditions such as Parkinson's disease, spasticity, and memory loss, as well as in physiological functions as diverse as hearing and the modulation of heart rate, where Ca$_V$1.3 channels play a key role.

Finally, our findings point to a novel, general mechanism for the dynamic modulation of ionic currents through L-type Ca$_V$1.3$_S$ and Ca$_V$1.2 channels. For example, in SA node cells, Ca²⁺ entry through Ca$_V$1.3 and Ca$_V$1.2 channels is a key feature in the generation of pacemaker activity (*Mangoni et al., 2003*; *Platzer et al., 2000*; *Striessnig et al., 2014*). In addition, it has been shown in a recent paper that Ca$_V$1.3 channels play a critical role in controlling pacemaker activity in SAN cells through the activation of RyR and the triggering of local Ca²⁺ release from the sarcoplasmic

reticulum (SR) (*Torrente et al., 2016*). The present results suggest a potential new mechanism for $Ca^{2+}$-dependent control of pacing in these cells. Accordingly, spontaneous junctional SR $Ca^{2+}$ release events could induce multimerization of nearby $Ca_V1.3_S$, which, once fused, could produce persistent inward $Ca^{2+}$ currents that increase the number of RyR activated, thereby driving SA node cells closer to the threshold for action potential generation.

## Materials and methods

### Plasmid constructs and tsA-201 cell transfection

The pcDNA clones of the rat $Ca_V1.3$ isoforms were obtained from Addgene. We used two Addgene $Ca_V1.3_S$ plasmids#26576 and #49333 (*Xu and Lipscombe, 2001*), the first one containing two single point substitutions, a glycine by a serine at position 244 and an alanine by a valine at position 1104. We found that they encoded channels with similar voltage-dependencies of activation and rate of activation (*Figure 5—figure supplement 2A and B*). The $Ca_V1.3_S$ channels encoded by these plasmids were also capable of undergoing $Ca^{2+}$-dependent dimerization and functional coupling (*Figure 5—figure supplement 2C–F*). On the basis of these data, we concluded that the G244S and A1104V substitutions in the plasmid #26576 are functionally silent with respect to voltage dependence, rate of activation and have no effect on the capacity of adjacent $Ca_V1.3_S$ channels to undergo allosteric interactions. A prior study by Lieb et al. also reported no contribution of these mutations to the functional properties of $Ca_V1.3_S$ channels (*Lieb et al., 2012*),. Both plasmids #26576 and #49333 were used to express and design the functional $Ca_V1.3_S$ constructs used in this study. $Ca_V1.3_L$ was also obtained from Addgene (plasmid #49332, (*Xu and Lipscombe, 2001*)). Auxiliary subunits $Ca_V\beta_3$ and $Ca_V\alpha_2\delta_1$ were gift of Dr. Diane Lipscombe's laboratory; Brown University, Providence, RI). The C-terminus of $Ca_V1.3_S$ and $Ca_V1.3_L$ channels were fused to different proteins depending on the experimental approach. For bimolecular fluorescence complementation, they were fused to either VN or VC fragments of the Venus protein (*Kodama and Hu, 2010*) (Dr. Chang-Deng Hu, Addgene plasmids 27097, 22011; for photobleaching experiments, they were fused to monomeric GFP ($mGFP_{A206K}$), amplified from the pCGFP-EU vector (*Kawate and Gouaux, 2006*), kindly provided by Dr. Eric Goaux (Oregon Health and Science University, Portland, OR); and for the light induced cryptochrome system, they were fused with either CRY2 or CIBN (generous gifts from Dr. Pietro Di Camilli, Yale University, New Haven, CT). The $CaM_{1234}$ plasmid was a generous gift from Dr. Johannes Hell (University of California, Davis, CA).

The tsA-201 cell line, used for heterologous expression of the constructs listed above, was maintained in Dulbecco's modified Eagle medium supplemented with 10% fetal bovine serum and 1% penicillin/streptomycin antibiotic solution. Cells were transiently transfected using jetPEI transfection reagent (Polyplus Transfection, New York, NY) and plated onto 25-mm coverslips (0.13–0.17-mm thick). Successfully transfected cells were identified on the basis of turbo red fluorescent protein (tRFP) fluorescence. Imaging and electrophysiology experiments were performed within 48 hr of transfection.

### Hippocampal neuron culture and transfection

Hippocampal neurons were prepared from newborn (P1) Sprague-Dawley rats in accordance with University of Washington (UW) guidelines. Animals were decapitated and their tissue harvested according to a protocol approved by the UW Institutional Animal Care and Use Committee (IACUC). The hippocampi of six rat pups were dissected and cut into small pieces in cold dissection medium consisting of 12 mM $MgSO_4$ and 0.3% bovine serum albumen (BSA) in Hank's balanced salt solution (HBSS). The pieces were incubated for 30 min at 37°C in dissection medium containing 25 U/ml papain. The digested tissue was washed with warm Neuronal medium consisting of Minimal Essential Medium (MEM) supplemented with 10% horse serum, 2% B27, 25 mM HEPES, 20 mM glucose, 2 mM GlutaMAX, 1 mM sodium pyruvate, and 1% penicillin/streptomycin antibiotic solution. The tissue was then gently homogenized in fresh Neuronal medium using a long Pasteur pipette. Neurons were plated on poly-D-lysine-coated coverslips (0.2 mg/ml for 2 hr) at a density of $2 \times 10^5$ cells/coverslip. After incubating neurons at 35°C for 24 hr, unattached cells were removed by replacing the medium with fresh Neuronal medium. Every 5 day, one-third of the medium was replaced with fresh

Neuronal medium supplemented with the anti-mitotic agents fluorodeoxyuridine (20 μM) and uridine (50 μM).

After 14 d in culture, rat hippocampal neurons were transfected using 2.4 μg DNA, 4.8 μl of Lipofectamine LTX and 4.8 μl PLUS reagent (Life Technologies, Grand Island, NY) in a final volume of 1 ml of a 1:1 mixture of Neuronal medium and Opti-MEM. After 1 hr of incubation, the medium was replaced with fresh Neuronal medium. Experiments were performed within 48 hr of transfection. Successful transfection was corroborated by detection of tRFP.

## Electrophysiology

$Ca^{2+}$ currents were recorded using the whole-cell configuration of the patch-clamp technique in voltage-clamp mode. Currents were sampled at a frequency of 10 kHz and low-pass filtered at 2 kHz using an Axopatch 200B amplifier. During the experiments, tsA-201 cells were superfused with a solution containing 5 mM CsCl, 10 mM HEPES, 10 mM glucose, 140 mM N-methyl-D-glucamine, 1 mM $MgCl_2$ and 2 mM $CaCl_2$ or 2 mM $BaCl_2$, depending on the experiment. pH was adjusted to 7.4 with HCl. For the experiments using 20 mM $CaCl_2$, the osmolarity was adjusted by decreasing the concentration of NMDG to 113 mM. Borosilicate patch pipettes with resistances of 3–6 MΩ were filled with an internal solution containing 87 mM cesium aspartate (CsAsp), 20 mM CsCl, 1 mM $MgCl_2$, 10 mM HEPES, 10 mM EGTA and 5 mM MgATP, adjusted to pH 7.2 with CsOH. A voltage offset of 10 mV, attributable to the liquid junction potential of these solutions, was corrected offline. Current–voltage relationships were obtained by subjecting cells to a series of 300-ms depolarizing pulses from a holding potential of -80 mV to test potentials ranging from -70 to +50 mV. The voltage dependence of channel activation ($G/G_{max}$) was obtained from the resultant currents by converting them to conductances using the equation, $G = I_{Ca}$/(test pulse potential – reversal potential of $I_{Ca}$); normalized $G/G_{max}$ was plotted as a function of test potential. We found that the reversal potential for $Ca_V1.3_L$ (+51 ± 4 mV) and $Ca_V1.3_S$ (+54 ± 3 mV) currents in the presence of 2 mM $Ca^{2+}$ were not significantly different (p=0.49). The same was true with 2 mM $Ba^{2+}$ in the bath ($Ca_V1.3_L$ = +60 ± 5 mV; $Ca_V1.3_S$ = +65 ± 2 mV; p=0.31). Thus, while $E_{rev}$ is similar in $Ca_V1.3_L$ and $Ca_V1.3_S$ channels with the same permeating ion, the reversal potential of $Ca^{2+}$ and $Ba^{2+}$ currents through these channels differ. In our patch clamp experiments in which the external solution was switched from $Ba^{2+}$ to $Ca^{2+}$, 2-min intervals were inserted between the onset of the whole cell configuration to the first pulse and again after switching the external solution, to rule out any effect of the run-up of the $I_{Ca}$ (*Tiaho et al., 1993*).

Single-channel currents ($i_{Ca}$) were recorded from tsA-201 using the cell-attached configuration. Cells were superfused with a high $K^+$ solution to fix the membrane potential at ~0 mV. The bathing solution had the following composition: 145 mM KCl, 2 mM $MgCl_2$, 0.1 mM $CaCl_2$, 10 mM HEPES and 10 mM glucose; pH was adjusted to 7.3 with KOH. Pipettes were filled with a solution containing 10 mM HEPES and either 20 mM $CaCl_2$ or 20 mM $BaCl_2$; pH was adjusted to 7.2 with CsOH. The dihydropyridine agonist BayK-8644 (500 nM) was included in the pipette solution to promote longer channel open times. A voltage-step protocol from a holding potential of −80 mV to a depolarized potential of −30 mV was used to elicit currents. The single-channel event-detection algorithm of pClamp 10.2 was used to measure single-channel opening amplitudes. We generated all-points histograms from our cell-attached patch-clamp recordings. These histograms were fit using Prism 5.0a software (GraphPad software Inc. La Jolla, CA) with a multi-Gaussian function that included a quantal (q; i.e., elementary current) parameter using the following equation:

$$N = \sum_{j=1}^{n} a_j \times e^{\left[-\frac{i_{Ca}-jq^2}{2jb}\right]},$$

where $N$ is the number of events, $a$ and $b$ are constants, $i_{Ca}$ is the amplitude of the current measured and $q$ is the quantal elementary current of the channel.

Spontaneous discharge of cultured hippocampal neurons was recorded in current-clamp mode using the perforated-patch configuration. Neurons were superfused with a solution containing 140 mM NaCl, 5 mM KCl, 10 mM HEPES, 10 mM glucose, 1 mM $MgCl_2$, 2 mM $CaCl_2$ and 1 mM Na-pyruvate, adjusted to pH 7.4 with NaOH. Borosilicate patch pipettes with resistances of 3–6 MΩ were filled with an internal solution containing 5 mM NaCl, 140 mM KCl, 15 mM HEPES and 7 mM MgATP, adjusted to pH 7.2 with KOH; 60 μM amphotericin B was added to the solution before

starting the recording. Series resistances lower than 30 MΩ were obtained within 5 min of seal formation. The sampling frequency was 10 kHz filtered at 2 kHz.

## Immunofluorescence and super-resolution microscopy

For immunostaining $Ca_V1.3$ in tsA-201 cells or hippocampal neurons, cells were fixed by incubating in phosphate-buffered saline (PBS) containing 3% paraformaldehyde and 0.1% glutaraldehyde for 15 min. After washing with PBS, cells were incubated with 50 mM glycine at 4°C for 10 min (aldehyde reduction), washed again with PBS, and blocked by incubating with 20% SEA BLOCK (Thermo Scientific) and 0.25% v/v Triton X-100 in PBS (blocking buffer) for 1 hr. The cells were incubated overnight at 4°C with primary antibodies recognizing the residues 809 to 825 located at the intracellular II-III loop of the $Ca_V1.3$ channel (DNKVTIDDYQEEAEDKD, rabbit; provided by Drs. William Catterall and Ruth Westenbroek) and the neuronal marker MAP2 (mouse; Abcam), diluted in blocking buffer to a concentration of 10 µg/ml. Cells were then washed with PBS, incubated for 1 hr with Alexa Fluor 647-conjugated donkey anti-rabbit (2 µg/ml; Molecular Probes) and Alexa Fluor 488-conjugated chicken anti-mouse (2 µg/ml; Molecular Probes) secondary antibodies, and washed again with PBS. Our antibody was designed to bind to the intracellular loop linking the 2nd and 3rd membrane domains of $Ca_V1.3$. It cannot distinguish between $Ca_V1.3_S$ and $Ca_V1.3_L$.

For super-resolution microscopy, coverslips were mounted on microscope slides with a round cavity using MEA-GLOX imaging buffer (NeoLab Migge Laborbedarf-Vertriebs GmbH, Germany) and sealed with Twinsil (Picodent, Germany). The imaging buffer contained 10 mM MEA, 0.56 mg/ml glucose oxidase, 34 µg/ml catalase, and 10% w/v glucose in TN buffer (50 mM Tris-HCl pH 8, 10 mM NaCl).

A super resolution ground-state depletion system (SR-GSD, Leica) based on stochastic single-molecule localization was used to generate super-resolution images of $Ca_V1.3$ in hippocampal neurons and tsA-201 cells. The Leica SR-GSD system was equipped with high-power lasers (488 nm, 1.4 kW/cm$^2$; 532 nm, 2.1 kW/cm$^2$; 642 nm, 2.1 kW/cm$^2$) and an additional 30 mW, 405 nm laser. Images were obtained using a 160× HCX Plan-Apochromat (NA 1.43) oil-immersion lens and an EMCCD camera (iXon3 897; Andor Technology). For all experiments, the camera was running in frame-transfer mode at a frame rate of 100 Hz (10 ms exposure time). Fluorescence was detected through Leica high-power TIRF filter cubes (488 HP-T, 532 HP-T, 642 HP-T) with emission band-pass filters of 505–605 nm, 550–650 nm, and 660–760 nm.

Super-resolution localization images of $Ca_V1.3$ channel distribution were reconstructed using the coordinates of centroids obtained by fitting single-molecule fluorescence signals with a 2D Gaussian function using LASAF software (Leica). A total of 50,000–100,000 images were used to construct the images. The localization accuracy of the system is limited by the statistical noise of photon counting. Thus, assuming the point-spread functions are Gaussian, the precision of localization is proportional to $DLR/\sqrt{N}$, where DLR is the diffraction-limited resolution of a fluorophore and N is the average number of detected photons per switching event (*Dempsey et al., 2011*; *Folling et al., 2008*). Accordingly, we estimated a lateral localization accuracy of 16 nm for Alexa 647 (~1900 detected photons per switching cycle). $Ca_V1.3$ cluster size was determined using binary masks of the images in ImageJ software (NIH).

## Step-wise photobleaching

The number of $Ca_V1.3_S$ channels in clusters along the surface membrane was estimated using a single-molecule bleaching approach similar to that described by *Ulbrich and Isacoff (2007)*. Briefly, TIRF images of tsA-201 or hippocampal neurons expressing $Ca_V1.3_S$ channels fused to the monomeric GFP were acquired using our Leica GSD microscope in TIRF mode. Cells were fixed with 4% paraformaldehyde for 10 min prior to the acquisition. Images were acquired using an oil immersion 160x objective (NA 1.43) and an Andor iXON EMCCD camera. GFP was excited with a 488 nm laser and image stacks of 2000 frames were acquired at 30 Hz. During analysis, the first 5 images of the stack were averaged and a rolling-ball background subtraction was applied using ImageJ (NIH). This image was then low-pass filtered with a 2 pixel cut-off and high-pass filtered with a 5 pixel cut-off. Thresholding was then applied to identify connected regions of pixels that were above threshold. The ImageJ plugin 'Time Series Analyzer v2.0' was used to select 4x4 pixel ROIs, centered on the

peak pixel in each spot. These ROIs were used to plot Z-axis intensity profiles (where z is time) of the entire image stack to manually detect the bleaching steps.

## Split Venus bi-molecular fluorescence complementation

The spontaneous interaction between the C-terminus of $Ca_V1.3_S$ channels was assessed using the split Venus system (*Shyu et al., 2006*). In this approach, $Ca_V1.3_S$ channels were fused to either the VN fragments ($N_{1–154}$) or the VC fragment ($C_{155–238}$) of the Venus fluorescent protein. The Venus protein emits fluorescence only when the two fragments are close enough to interact and reconstitute the whole protein, providing a measure of the proximity between the C-terminus of the $Ca_V1.3_S$ channels. $Ca_V1.3_S$-VN and/or $Ca_V1.3_S$-VC constructs were expressed at a 1:1 ratio in hippocampal neurons; tRFP fluorescence was used as an indicator of successful transfection. Confocal images were acquired with a Fluoview FV1000 microscope (Olympus, Center Valley, PA) equipped with a UPlanS-Apochromat 60× (NA 1.2) water-immersion objective. The Venus protein was excited using a 488 nm laser line. The calcium dependence of $Ca_V1.3$ spontaneous interactions was studied in tsA-201 cells expressing $Ca_V1.3_S$-VC alone or $Ca_V1.3_S$-VN and $Ca_V1.3_S$-VC in a 1:1 ratio. Using the whole-cell configuration of the patch-clamp technique, cells were depolarized from a holding voltage of -80 mV to test potentials ranging from -60 to +60 mV, administered as 9-s pulses. Maturation of newly reconstituted Venus protein takes some time, hence the long depolarizing pulse (*Nagai et al., 2002*). Images were acquired at a frequency of 100 Hz during each depolarizing pulse using a through-the-lens TIRF microscope built around an inverted microscope (IX-70; Olympus) equipped with a Plan-Apochromat (60×; NA 1.49) oil-immersion lens (Olympus) and an electron-multiplying charge-coupled device (EMCCD) camera (iXON; Andor Technology, UK). The last 10 images of each stack were averaged, and total fluorescence was quantified using ImageJ software (NIH). The images were pseudo-colored using the 'red hot' lookup table in ImageJ. $F_0$ was calculated by dividing the total fluorescence for each voltage by the initial fluorescence at -80 mV. The change in $F/F_0$ was plotted against voltage and compared to $G/G_{max}$ curves constructed as described in the electrophysiology section. The $Ca^{2+}$ dependence of Venus reconstitution was tested in cells bathed in an external solution containing 20 mM $Ca^{2+}$ or 2 mM $Ba^{2+}$.

## $Ca_V1.3_S$ sparklet imaging and analysis

$Ca_V1.3_S$ sparklets were recorded using the TIRF microscopy system described above. $[Ca^{2+}]_i$ was monitored by adding the $Ca^{2+}$ indicator Rhod-2 (200 µM) to the pipette solution and exciting with a 568 nm laser. The much slower $Ca^{2+}$ buffer EGTA (10 mM) was included with the relatively fast $Ca^{2+}$ indicator, Rhod-2, to restrict $Ca^{2+}$ signals to the vicinity of the $Ca^{2+}$ entry source. Sparklets were detected in tsA-201 cells expressing $Ca_V1.3_S$ channels. The driving force for $Ca^{2+}$ entry was increased by holding the membrane potential at -80 mV using the whole-cell configuration of the patch-clamp technique.

TIRF images were acquired at a frequency of 100 Hz using TILL Image software. Sparklets were detected and analyzed using custom software written in MATLAB (*Source code 1*). Fluorescence intensity values were converted to nanomolar units as described previously (*Navedo et al., 2007*). Event amplitude histograms were generated from $[Ca^{2+}]_i$ records and fitted with a multicomponent Gaussian function. We determined the activity of sparklets by calculating the $nP_s$ of each sparklet site, where $n$ is the number of quantal levels and $P_s$ is the probability that a quantal sparklet event is active. A detailed description of this analysis can be found in Navedo et al. (*Navedo et al., 2006*).

In split Venus experiments, sparklets were acquired at -80 mV before and after a depolarizing protocol from -60 to +60 mV with 9-s pulses. Sparklet images were always acquired in a solution containing 20 mM $Ca^{2+}$, whereas depolarization protocols were run in 20 mM $Ca^{2+}$ or 2 mM $Ba^{2+}$ to compare the effect of $Ca^{2+}$-dependent $Ca_V1.3_S$ dimerization on sparklet activity.

The degree of coupling between $Ca_V1.3$ $Ca^{2+}$ sparklet sites was assessed by further analyzing sparklet recordings using a binary coupled Markov chain model, as first described by *Chung and Kennedy (1996)* and previously employed by our group (*Navedo et al., 2010*; *Cheng et al., 2011*; *Dixon et al., 2012*). The custom program (*Source code 2*), written in the MATLAB language, assigns a coupling-coefficient ($\kappa$) to each record, where $\kappa$ can range from 0 (purely independently gating channels) to 1 (fully coupled channels). Elementary event amplitudes were set at 38 nM.

## Cryptochrome light-induced Ca$_V$1.3 dimerization

Light-induced dimerization of Ca$_V$1.3 channels was accomplished by fusing Ca$_V$1.3 channels with the optogenetic light-induced dimerization system based on CRY2 and CIB1 proteins of *Arabidopsis thaliana* (*Kennedy et al., 2010*). Upon blue-light illumination (488 nm), CRY2 absorbs a photon, causing a conformational change in one of its domains that promotes binding to the N-terminal region of CIB1 (CIBN). Ca$_V$1.3-CIBN and/or Ca$_V$1.3-CRY2 constructs were expressed in a 1:1 ratio in hippocampal neurons. Forty-eight hours after transfection, spontaneous action potential firing was recorded in current-clamp mode. One minute after initiating recordings, neurons were exposed to a 30-s blue light pulse to induce dimerization, and the changes in the firing pattern were measured. In tsA-201 cells, Ca$^{2+}$ currents were recorded before and after light illumination in response to a 20-ms depolarizing pulse at +10 mV from a holding potential of -80 mV. Experiments were performed on a Nikon (Eclipse TE2000-S) Swept Field confocal system controlled with Elements software and equipped with a 488 nm laser line and a Plan Apo 60× 1.45 N.A. oil-immersion objective.

## Data analysis

Data were collected from at least five independent experiments in each series. The data included in this paper were normally distributed. Accordingly, parametric statistics were performed and mean ± SEM are used to provide a description of the data set. Student's t-test was used to test for statistical significance using Prism 5.0 a software (GraphPad software Inc. La Jolla, CA). We decided, *a priori*, that p values <0.05 were indicative a statistical significance difference between or among data groups. The number of cells used for each experiment and *p* values are detailed in each figure legend. Paired t- tests were used to test for statistical significance of paired observations. Comparisons between three or more conditions were made by one-way ANOVA test using Prism software.

## Acknowledgements

We thank Dr. Bertil Hille and Dr. Manuel F Navedo for comments and Dr. Joshua Vaughan for help with super-resolution analysis. Dr. William Catterall and Dr. Ruth E Westenbroek provided antibodies against Ca$_V$1.3.

## Additional information

### Funding

| Funder | Grant reference number | Author |
|---|---|---|
| National Heart, Lung, and Blood Institute | HL085870 | Luis F Santana |
| National Heart, Lung, and Blood Institute | HL085686 | Luis F Santana |
| National Institute of Neurological Disorders and Stroke | NS077863 | Marc D Binder |
| American Heart Association | 15SDG25560035 | Rose E Dixon |

The funders had no role in study design, data collection and interpretation, or the decision to submit the work for publication.

### Author contributions

CMM, RED, Conception and design, Acquisition of data, Analysis and interpretation of data, Drafting or revising the article; ST, Acquisition of data, Analysis and interpretation of data; CY, XO-A, Acquisition of data, Analysis and interpretation of data, Contributed unpublished essential data or reagents; MDB, Conception and design, Drafting or revising the article; LFS, Conception and design, Analysis and interpretation of data, Drafting or revising the article

### Author ORCIDs

Luis F Santana, http://orcid.org/0000-0002-4297-8029

## Ethics

Animal experimentation: This study was performed in strict accordance with the recommendations in the Guide for the Care and Use of Laboratory Animals of the National Institutes of Health. All of the animals were handled according to approved institutional animal care and use committee (IACUC) protocols (#3374-01) of the University of Washington and (#18896) of the University of California, Davis.

## Additional files

### Supplementary files

- Source code 1. Custom software for Ca2+ sparklet detection and analysis written in MATLAB.
- Source code 2. Binary coupled Markov chain model.

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
