## [Decision Letter]

Thank you for resubmitting your work entitled "Ca^2+^ entry into neurons is facilitated by cooperative gating of clustered Ca_V_1.3 channels" for consideration at *eLife*. Your article has been favorably evaluated by Richard Aldrich (Senior editor and reviewing editor) and two reviewers: Jorg Stressing and Henry Colecraft. The manuscript has been improved but there are some remaining issues that need to be addressed before acceptance, as outlined below:

The required revisions should be easily accomplished and are detailed in the individual reviews below.

*Reviewer #1:*

The authors have done a remarkable job in revising their manuscript. There are only a few issues remaining:

Introduction: As mentioned in my previous comments, the Chan et al. paper should not be cited in the context of pacemaking in dopamine neurons because their role for pacemaking as described in this paper has been corrected in a second paper by the same authors. In the first paragraph it therefore would make sense to replace the Chan et al. reference by their Guzman et al., 2009 reference. In the third paragraph "drive the" should be replaced by "support". Also here the Chan reference should be replaced by Guzman.

In the third paragraph of the Introduction and in the first paragraph of the subsection “Coupling of Ca_V_1.3_S_ channels increases the firing rate of hippocampal neurons”: The estimated fraction of Ca_V_1.3 channels in hippocampal neurons is 20%: total 22.3% of current; 4.4% remaining in Ca_V_1.2 knockout (Moosmang paper).

In the second paragraph of the subsection “A Ca^2+^-dependent mechanism mediates facilitation of Ca_V_1.3_S_, but not Ca_V_1.3_L_, channels”: The authors should somehow make clear what leads them to conclude opening of 2 and more channels. I assume they derive this interpretation from the double (multiple) amplitudes of unitary currents. However, it also could be that smaller unitary currents represent subconductance states of a channel. A clarifying sentence would make sense here.

Also: "the frequency of currents…is higher…(Figure 1)”. How is this difference between the splice variants quantified ("higher"); statistics (if "higher" is significant) should be provided here.

In the third paragraph of the subsection “A Ca^2+^-dependent mechanism mediates facilitation of Ca_V_1.3_S_, but not Ca_V_1.3_L_, channels”: "We found that Ca^2+^ ions entering the cell through the channels increased the NPo nearly 1.5-fold for Ca_V_1.3_S_ channels". I assume the authors have done the controls showing that run-up during the experiment and during switching from equilibrium in Ba- to Ca-containing solutions cannot explain the increased Ca_V_1.3_S_ current. This information should be provided (e.g. in the Methods section).

Based on their experimental conditions channels and channel clusters (fluorescence or antibody stained) they analyze in tsA-201 cells and neurons must reflect both intra- (unlikely functional) and plasmalemmal (surface, i.e. functional) channel complexes. In the absence of other evidence this limitation should be mentioned in the Results section.

In the subsection “Physical interaction between Ca_V_1.3_S_ channels induces *I_Ca_* facilitation”: I assume paired t-test or Wilcoxon test has been used for calculating statistics of paired observations. Use of this test should be indicated also in the statistics section of the Methods part.

In the fifth paragraph of the Discussion: Also long Ca_V_1.3 splice variants activate more negative that any Ca_V_1.2 splice variant characterized so far. Therefore it would be clearer to state, e.g.: Although Ca_V_1.3 channels are generally classified as high-voltage activated channels, their lower activation range allows them to generate subthreshold depolarizations that support repetitive firing (Olson et al., 2005). This is in particular true for short splice variants which are even more voltage-sensitive than Ca_V_1.3L (Bock et al., 2011; Tan et al., 2001-PMID 21998309).

Note that the voltage-sensitivity is changed between long and short forms, i.e. the coupling of voltage-sensor movements to channel opening (Lieb et al., 2014; C-terminal modulatory domain controls coupling of voltage-sensing to pore opening in Ca_V_1.3 L-type Ca(2+) channels).

Please replace Xu and Lipscombe here. These authors actually generated the corrupted Ca_V_1.3_L_ variant that was originally used in this paper. Therefore they could never detect the difference between Ca_V_1.3_S_ and Ca_V_1.3_L_. Cite the Soong group here instead.

In the subsection “Plasmid constructs and tsA-201 cell transfection”: The #26576 must differ from the #49333 plasmid not only at position 244 (G244S) but also should contain a non-natural substitution at position 1104 (A1104V). This is not discussed by the authors. However, this is as relevant as G244S for interpreting their results. However, the effects of these substitutions have been characterized in detail in an earlier publication (Lieb et al., 2012; PMID 22760075). Based on these findings a contribution of this substitution to the function of Ca_V_1.3 is unlikely. If this substitution is indeed present in their cDNA they should mention this and cite the above paper in the Methods part.

New Ca_V_1.3_L_ experiments have been performed months after most of the Ca_V_1.3_S_ experiments shown. Have the authors also confirmed the persistence of the differences in parallel experiments with Ca_V_1.3_S_ when new Ca_V_1.3_L_ data were generated?

*Reviewer #2:*

The authors have been responsive to the previous critiques, and the manuscript is improved. Overall, the work is pretty convincing and I believe that it makes a valuable contribution to the Ca_V_ channel literature. There are still a few points that need further clarification.

1) Results: "Interestingly, we found that the frequency of currents resulting from the simultaneous opening of 2 or more channels is higher for Ca_V_1.3_S_ than Ca_V_1.3_L_ channels when Ca^2+^ is the charge carrier (Figure 1)."

This certainly seemed the case in the first version of the manuscript but is not obvious to me from the data provided in the revised manuscript. The authors should either provide more quantitative information to back up this statement or remove it.

2) Results: "This assumption is reasonable, as we found no significant difference in the activation time constants for Ca_V_1.3_L_ and Ca_V_1.3_S_ currents that were 1.60 ± 0.22 and 1.17 ± 0.051 ms (-10 mv), respectively."

Please provide n and P values for the statistical analyses.

3) Sparklet data shown in Figure 2, Figure 5 and Figure 6 have not changed from previous version, yet now they are labeled as being recorded at -80 mV whereas the original version had them recorded at -70 mV. Which one is correct?

4) Results: "We found that, with 2 mM Ca^2+^ in the bathing solution, cells expressing Ca_V_1.3_L_Δ116 channels failed to reconstitute Venus fluorescence, similar to what we observed for the full length Ca_V_1.3_L_ channel (Figure 8)."

Experiment should be done with 20 mM Ca^2+^ to mimic conditions used for Ca_V_1.3_S_ in Figure 5and Figure 6. The failure to observe Ca_V_1.3_L_Δ116 reconstitution of Venus fluorescence could be simply due to the 10-fold less extracellular Ca^2+^ used in this experiment. If so, this would change the conclusion reached for this experiment.

5) Discussion: "An intriguing observation in our study is that physical Ca_V_1.3_S_ channel coupling (assessed by Venus reconstitution) and the normalized conductance of *I_Ca_* (i.e., G/G_max_) have similar voltage dependencies. We propose that this is a consequence of two factors. First, Venus fluorescence (i.e., reconstitution) and Ca_V_1.3_S_ conductance have similar sigmoidal relationships suggesting that Ca_V_1.3_S_ coupling is primarily determined by the open probability of these channels. This is consistent with our model, as the extent of Venus reconstitution depends on the local calcium concentration produced by the coordinated opening of clustered channels and less on magnitude of the flux through individual channels. The second factor contributing to the shared voltage dependencies is the irreversible nature of the Venus reconstitution, resulting in an increased number of fluorescent proteins with each successive depolarization as the open probability of Ca_V_1.3_S_ channels increases."

These lines should be removed as they represent an over-interpretation of the data. As the authors note in their second point, the sigmoidal nature of the Venus reconstitution dependence on voltage most likely reflects the irreversible nature of the Venus reconstitution such that fluorescence accumulates with successive depolarizations. I suspect if the experiments were done in reverse (i.e., test pulses stepping down from +60 mV to -60 mV) that the relationship would also show a sigmoidal relationship but with the fluorescence lower at the higher voltages where channel Po is higher. This thought experiment suggests the idea that "…Ca_V_1.3_S_ coupling is primarily determined by the open probability of these channels…" is an oversimplification.

6) Please include the procedures for single-channel recordings in the Methods. The single channel traces appear to have quite long openings (particularly for the Ba^2+^ traces; Figure 1—figure supplement 1). Was an L-type channel agonist included for these experiments? If so this should be explicitly stated in the Methods.

[Editors’ note: a previous version of this study was rejected after peer review, but the authors submitted for reconsideration. The previous decision letter after peer review is shown below.]

Thank you for choosing to send your work entitled "Ca^2+^ entry into neurons is facilitated by cooperative gating of clustered Ca_V_1.3 channels" for consideration at *eLife*. Your full submission has been evaluated by Eve Marder (Senior editor), Richard Aldrich (Reviewing editor), and two peer reviewers, one of whom, Jorg Striessnig, has agreed to reveal his identity. The decision was reached after discussions between the reviewers. Based on our discussions and the individual reviews below, we regret to inform you that your work will not be considered further for publication in *eLife*.

While we are enthusiastic about the work, significant concerns about the appropriateness of the particular constructs used, as detailed in the enclosed reviews, preclude us from accepting the manuscript, or asking for a straightforward revision. *eLife*'s policy is to reject papers rather than ask for extensive new work. However, if you do the work with the appropriate constructs, we would welcome a new submission. In that case, please indicate in your cover letter that you would like the same BRE member and potentially reviewers (although we can make no promises) to look at it. Of course, if you do not agree with the reviewers, you are welcome to submit this manuscript elsewhere, as it is no longer in consideration at *eLife*.

*Reviewer #1:*

This paper follows up on intriguing work from the Santana lab suggesting that Ca^2+^ influx through L-type Ca_V_1.2 channels promotes a Ca^2+^-CaM dependent physical coupling of two or more channels through the C-termini of the pore-forming alpha1C subunit. This physical coupling leads to co-operative opening of the channels and facilitates Ca^2+^ influx. Here they extend this analyses to Ca_V_1.3 channels and observe the same phenomenon in a Ca_V_1.3 short but not Ca_V_1.3 long splice variant. They further suggest that this phenomenon occurs and contributes to excitability in hippocampal neurons.

The work is impressive in the amount of different techniques and experiments the authors bring to bear on the question at hand. If correct, this would certainly be an important contribution to the literature on gating mechanisms and physiological regulation of L-type channels. However, in its current form the work has a number of inconsistencies and issues with interpretation of the data that significantly weaken the strength of the overall conclusions. Given that this work adheres closely to previously published work on Ca_V_1.2 it is my opinion that the presentation here should go substantially beyond the phenomenology of the observation and provide a more rigorous mechanism for the effect.

1) Single channel traces with Ca^2+^ as charge carrier show no signature of Ca^2+^-dependent inactivation despite this being so prominent in whole-cell currents (Figure 1). Why is this? The authors should provide ensemble averages of the single channel data to increase confidence that the unitary currents actually correspond to Ca_V_1.3.

2) A central point of the paper is that Ca_V_1.3s but not Ca_V_1.3_L_ displays the phenomenon of Ca^2+^-induced channel coupling and facilitation. Ultimately, this is attributed the distal C-terminus competing away CaM binding to the channel. However, the Ca_V_1.3_L_ used in the paper demonstrates quite a robust CDI (e.g. Figure 1), indicating that CaM is bound to the channel. The comparable CDI between Ca_V_1.3s and Ca_V_1.3_L_ observed in the paper is likely due to the use of the rat clones which are known to display similar CDI at ambient CaM concentrations (Yang et al., 2006, J. Neurosci. 26:10677-89). This is because the rat Ca_V_1.3_L_ has a point mutation relative to the human clone, which weakens its affinity for the CaM-binding domain such that ambient CaM is sufficient to (by competition) access its site on the channel (Liu et al., 2010, Nature, 463:968-973). Given this, it is unclear why Ca_V_1.3_L_ would lack the Ca^2+^-dependent facilitation described here. At the very least the interpretation that this due to CaM being excluded from these channels is not supported?

3) What is the kinetics of the facilitation? Figure 1 suggest it occurs immediately upon channel activation within 1 ms. This seems very fast for the inter-molecular binding mechanism proposed in Figure 10. Also, how fast does the system reset after closing the channel?

4) Does CaM_1234_ prevent the facilitation observed with Ca_V_1.3s in whole-cell currents (i.e. what does Figure 1 look like with CaM_1234_)?

5) Light-induced dimerization experiments in Figure 4. Seems to me that these experiments should be done with Ba^2+^ and not Ca^2+^ as charge carrier? If idea is the dimerization is sufficient to induce facilitation, having Ca^2+^ also present complicates the interpretation.

6) What is the CaM that actually produces the proposed channel coupling? The ability of CaM_1234_ to prevent facilitation would suggest that apoCaM preassociated with the channel is responsible. However, the effectiveness of MLCKp suggests a soluble CaM. If both of these are true then it is inconsistent with the simple model proposed in Figure 10. The authors need to provide a model that fits all the facts they present.

7) Figure 6: Is the increase in firing rate decreased by L-type channel antagonists?

*Reviewer #2:*

In their manuscript, Moreno and colleagues extend their previous finding (published in *eLife*) of cooperative gating of Ca_V_1.2 voltage gated Ca^2+^ channels to structurally and functionally highly related Ca_V_1.3 channels. As for Ca_V_1.2 in cardiomyocytes they made a big effort in using a battery of experimental paradigms to demonstrate calmodulin-dependent physical coupling leading to higher open probabilities in tsA-201 cells and hippocampal neurons. Their findings contrast with the current view of independent gating of Ca_V_-channels. By discovering a novel modulatory mechanism this paper is potentially very important to advance calcium channel research.

The paper is clearly written, well-illustrated and of appropriate length.

Two major points need to be considered to fully support the conclusions drawn from the experiments presented:

1) The authors describe the use of the Ca_V_1.3 alpha1 rat constructs Addgene plasmid 26577 (long) and 26576 (short). This selection is very unfortunate. Both constructs contain a number of mutations originally introduced by cloning the full-length alpha1-subunits. Addgene (meanwhile?) also provides the corrected versions (plasmids 49332 and 49333).

Unfortunately, the mutations are not functionally silent. The Yue group (Liu et al. Cell, 2010) reported that one of the mutations (Val to Ala in ICDI) disrupts the function of a distal modulatory domain (ICDI, also termed DCRD; a finding later confirmed by others: Huang et al. Mol Pharmacol, 2013; Lieb et al., Biophys J 2014). Since this domain competitively tunes the interaction of calmodulin with its upstream binding sites (pre-IQ, IQ) the long mutant Ca_V_1.3 channel behaves functionally more like short splice variants (that lack ICDI due to splicing), with higher CDI and higher open probability, properties that are crucial for the interpretation of the author's data. The authors should have been aware of this because the difference between Ca^2+^ current inactivation of their Ca_V_1.3L and Ca_V_1.3_S_ and between Ca_V_1.3_L_ and Ca_V_1.3delta116 was much smaller than expected between rat (and human) Ca_V_1.3_L_ and Ca_V_1.3_S_ variants (shown in many publications e.g. by the groups of Yue, Soong, and Striessnig).

To get around this problem, at least some key experiments affected by the use of mutant constructs should be repeated with their corrected counterparts, in particular those involving Ca^2+^ signals generated by the long isoform (Figure 1., Figure 2, Figure 4). This is justified due to the lower open probability and less pronounced CDI expected for the corrected (wildtype) rat Ca_V_1.3_L_ construct. Although there are also good arguments to repeat all other experiments with the correct constructs, the authors should be able to provide convincing arguments that this is not necessary, e.g. by discussing that CDI does not seem to be a determinant for coupling (Figure 7).

2) Evidence for the specificity of staining of endogenous Ca_V_1.3 alpha1 subunits in hippocampal neurons should be provided. So far no Ca_V_1.3 antibody staining has been reported in hippocampal neurons or brain sections that was shown to be absent in Ca_V_1.3 knockout controls. I would therefore not be surprised if this is also true for their antibody. If there is no evidence for such convincing specificity, the authors should either remove data showing staining of native channels or clearly state this caveat in their paper. Obviously, they cannot be blamed for the fact that no anti-Ca_V_1.3 antibody with proven specificity exists. In any case, they should show the typical somatodendritic clusters also on spines of their Ca_V_1.3 constructs transfected in hippocampal cultures using high resolution light microscopy as has been shown in previous work (e.g. Jenkins et al., J Neurosci, 2010).

3) In the subsection “Coupling of Ca_V_1.3_S_ channels increases the firing rate of hippocampal neurons”: According to Moosmang et al. (2005; PMID: 16251435), Ca_V_1.3 current components are minimal in hippocampal neurons and L-type currents seem to comprise only a small fraction of total voltage-gated calcium current. So how do the authors know their sparklets are produced by L-type currents? If so, they should vanish after treatment with a dihydropyridine Ca^2+^ channel blocker. This important control appears to be missing in the manuscript. They should also consider that there is no way to distinguish between Ca_V_1.3 and Ca_V_1.2 sparklets (not even by membrane voltage as stated in their manuscript).

---

## [Author Response]

Reviewer #1:

*The authors have done a remarkable job in revising their manuscript. There are only a few issues remaining: Introduction: As mentioned in my previous comments, the Chan et al. paper should not be cited in the context of pacemaking in dopamine neurons because their role for pacemaking as described in this paper has been corrected in a second paper by the same authors. In the first paragraph it therefore would make sense to replace the Chan et al. reference by their Guzman et al., 2009 reference. In the third paragraph "drive the" should be replaced by "support". Also here the Chan reference should be replaced by Guzman.* We thank the reviewer for the clarification. As suggested, we replaced the references where indicated.

*In the third paragraph of the Introduction and in the first paragraph of the subsection “Coupling of Ca_V_1.3_S_ channels increases the firing rate of hippocampal neurons”: The estimated fraction of Ca_V_1.3 channels in hippocampal neurons is 20%: total 22.3% of current; 4.4% remaining in Ca_V_1.2 knockout (Moosmang paper).*We thank the reviewer for the comment and have added that information to the manuscript (Introduction, third paragraph).

*In the second paragraph of the subsection “A Ca^2+^-dependent mechanism mediates facilitation of Ca_V_1.3_S_, but not Ca_V_1.3_L_, channels”: The authors should somehow make clear what leads them to conclude opening of 2 and more channels. I assume they derive this interpretation from the double (multiple) amplitudes of unitary currents. However, it also could be that smaller unitary currents represent subconductance states of a channel. A clarifying sentence would make sense here.*

We recognize that this analysis was not clearly described in the Methods section of the previous version of the manuscript. Briefly, we generated all-points histograms from our cell-attached patch-clamp recordings. These histograms were fit with a multi-Gaussian function that included a quantal (*q;* i.e., elementary current) parameter. This equation is included in the Methods section (subsection “Electrophysiology”) and the parameters used are included in the legend of Figure 1.

In the presence of 20 mM Ca^2+^, our analysis revealed that the amplitude of the elementary current (i.e., *q*) was -0.48 ± 0.07 pA and -0.49 ± 0.01 pA for Ca_V_1.3_L_ and Ca_V_1.3_S_ channels at -30 mV, respectively. Our *q_Ca_* is similar to previously reported values (Guia et al., 2001). This is now clarified in the Results. During analysis, the number of channel opening associated with a cell-attached current would be equal to the current amplitude divided by the corresponding *q*value (i.e., *q_Ca_* ± SD).

We recognize that Ca_V_1.3_S_ and Ca_V_1.3_L_ channels could potentially open in sub-conductance states (McDonald et al., 1994). However, we did not observe intermediate peaks between the current levels expected for sets of fully conducting channels. Thus, we concluded that the values reported correspond to the quantal values of the unitary current rather than to any sub-conductance state and that the double and multiple amplitudes of the currents correspond to the simultaneous opening of 2 or more channels.

*Also: "the frequency of currents…is higher…(Figure 1)”. How is this difference between the splice variants quantified ("higher"); statistics (if "higher" is significant) should be provided here.*We thank Reviewers 1 and 2 for pointing this out. Reviewer 2 (see below) is correct that the difference between the all-points Ca_V_1.3_S_ and Ca_V_1.3_L_ histograms was larger with the data included in the previous version of the manuscript. In this revision, we improved the fit of the multi-Gaussian function to the new Ca_V_1.3_S_ data (i.e., better goodness of fit). Although this analysis is consistent with the view that the frequency of currents resulting from the simultaneous opening of 2 or more channels is higher for Ca_V_1.3_S_ than Ca_V_1.3_L_ channels, we believe that additional work is necessary to carefully investigate the degree of coupling between Ca_V_1.3_S_ channels. Thus, as suggested by the reviewers, we scaled back the conclusions of these experiments to the following:

“These data suggest that the difference observed in the macroscopic currents between the Ca_V_1.3_L_ and Ca_V_1.3_S_ channels is not due to differences in unitary currents”.

*In the third paragraph of the subsection “A Ca^2+^-dependent mechanism mediates facilitation of Ca_V_1.3_S_, but not Ca_V_1.3_L_, channels”: "We found that Ca^2+^ ions entering the cell through the channels increased the NPo nearly 1.5-fold for Ca_V_1.3_S_ channels". I assume the authors have done the controls showing that run-up during the experiment and during switching from equilibrium in Ba- to Ca-containing solutions cannot explain the increased Ca_V_1.3_S_ current. This information should be provided (e.g. in the Methods section).* The reviewer raises an important point regarding the ‘run-up’ of Ca^2+^ currents. L-type Ca^2+^ currents typically reach their run-up peak after 2 minutes (Tiaho et al., 1993). In the patch-clamp experiments in which the external solution was switched from Ba^2+^ to Ca^2+^, 2-minute intervals were inserted between the onset of the whole cell configuration to the first pulse and again after switching the external solution to rule out any effect of the run-up of the *I_Ca_*. This is now clarified in the Methods section of the paper (subsection “Electrophysiology”, first paragraph).

*Based on their experimental conditions channels and channel clusters (fluorescence or antibody stained) they analyze in tsA-201 cells and neurons must reflect both intra- (unlikely functional) and plasmalemmal (surface, i.e. functional) channel complexes. In the absence of other evidence this limitation should be mentioned in the Results section.* For both the super-resolution and photobleaching experiments, we used TIRF microscopy with a penetration depth of about 130 nm to restrict imaging to a thin optical section near the plasma membrane. This is now mentioned in the revised manuscript (subsection “Ca_V_1.3 channels in hippocampal neurons aggregate in dense clusters”, third paragraph).

*In the subsection “Physical interaction between Ca_V_1.3_S_ channels induces I_Ca_ facilitation”: I assume paired t-test or Wilcoxon test has been used for calculating statistics of paired observations. Use of this test should be indicated also in the statistics section of the Methods part.* A paired t-test was indeed used to test for statistical significance of paired observations as now noted in the Data analysis section.

*In the fifth paragraph of the Discussion: Also long Ca_V_1.3 splice variants activate more negative that any Ca_V_1.2 splice variant characterized so far. Therefore it would be clearer to state, e.g.: Although Ca_V_1.3 channels are generally classified as high-voltage activated channels, their lower activation range allows them to generate subthreshold depolarizations that support repetitive firing (Olson et al., 2005). This is in particular true for short splice variants which are even more voltage-sensitive than Ca_V_1.3L (Bock et al., 2011; Tan et al., 2001-PMID 21998309).*

*Note that the voltage-sensitivity is changed between long and short forms, i.e. the coupling of voltage-sensor movements to channel opening (Lieb et al., 2014; C-terminal modulatory domain controls coupling of voltage-sensing to pore opening in Ca_V_1.3 L-type Ca(2+) channels). Please replace Xu and Lipscombe here. These authors actually generated the corrupted Ca_V_1.3_L_ variant that was originally used in this paper. Therefore they could never detect the difference between Ca_V_1.3_S_ and Ca_V_1.3_L_. Cite the Soong group here instead.* We agree with the reviewer that Ca_V_1.3_L_ also activates at more negative voltages than Ca_V_1.2 and that we need to clarify that the short variant Ca_v_1.3_S_channelsare even more voltage sensitive. We have modified the text and the references as suggested by the reviewer (Discussion, fifth paragraph).

*In the subsection “Plasmid constructs and tsA-201 cell transfection”: The #26576 must differ from the #49333 plasmid not only at position 244 (G244S) but also should contain a non-natural substitution at position 1104 (A1104V). This is not discussed by the authors. However, this is as relevant as G244S for interpreting their results. However, the effects of these substitutions have been characterized in detail in an earlier publication (Lieb et al., 2012; PMID 22760075). Based on these findings a contribution of this substitution to the function of Ca_V_1.3 is unlikely. If this substitution is indeed present in their cDNA they should mention this and cite the above paper in the Methods part.* We agree. The plasmid we used contained the two mutations as is now stated in the Methods section (subsection “Plasmid constructs and tsA-201 cell transfection”). In Figure 5—figure supplement 2 and the corresponding legend, Ca_V_1.3_S(G244S)_ has been replaced by (Ca_V_1.3_S(G244S/A1104V)_). We have also noted that a previous study by Lieb et al. (Lieb et al., 2012), found that these mutations did not alter the function of Ca_V_1.3_S_ channels (Line 555).

*New Ca_V_1.3_L_ experiments have been performed months after most of the Ca_V_1.3_S_ experiments shown. Have the authors also confirmed the persistence of the differences in parallel experiments with Ca_V_1.3_S_ when new Ca_V_1.3_L_ data were generated?* All of the additional experiments performed with the new Ca_V_1.3_L_ were complemented by parallel experiments on the previously used Ca_V_1.3_S_ channels to confirm the absence of coupling for the Ca_V_1.3_L_. In addition, split Venus and sparklets experiments for the Ca_V_1.3_L_ were run in parallel with those for the new Ca_V_1.3_S_ (Data for the short isoform is presented on Figure 5—figure supplement 2). Further, parallel single channel experiments, whole-cell *I_Ca_* and *NP_o_* analysis were also performed for the new Ca_V_1.3_L_ and Ca_V_1.3_S_ channels.

Reviewer #2:

*The authors have been responsive to the previous critiques, and the manuscript is improved. Overall, the work is pretty convincing and I believe that it makes a valuable contribution to the Ca_V_ channel literature. There are still a few points that need further clarification. 1) Results: "Interestingly, we found that the frequency of currents resulting from the simultaneous opening of 2 or more channels is higher for Ca_V_1.3_S_ than Ca_V_1.3_L_ channels when Ca^2+^ is the charge carrier (Figure 1)." This certainly seemed the case in the first version of the manuscript but is not obvious to me from the data provided in the revised manuscript. The authors should either provide more quantitative information to back up this statement or remove it.*

Please see our response above (“We thank Reviewers 1 and 2 for pointing this out […]).

*2) Results: "This assumption is reasonable, as we found no significant difference in the activation time constants for Ca_V_1.3_L_ and Ca_V_1.3_S_ currents that were 1.60 ± 0.22 and 1.17 ± 0.051 ms (-10 mv), respectively." Please provide n and P values for the statistical analyses.*

We had a sample size of 6 for each channel in these experiments. The *p* value was 0.112. This information has been added to the text (subsection “A Ca^2+^-dependent mechanism mediates facilitation of Ca_V_1.3_S_, but not Ca_V_1.3_L_, channels”, last paragraph).

*3) Sparklet data shown in Figure 2, Figure 5 and Figure 6 have not changed from previous version, yet now they are labeled as being recorded at -80 mV whereas the original version had them recorded at -70 mV. Which one is correct?* We thank the reviewer for the comment. -80 mV is the correct voltage. In the revised version of the manuscript, we have corrected the values for a liquid junction potential offset of 10 mV. On the previous version, we had not done so which is why all the sparklet figures were labeled with -70 mV. As all of the electrophysiology data presented were corrected for the LJP, we decided to correct sparklet data as well.

*4) Results: "We found that, with 2 mM Ca^2+^ in the bathing solution, cells expressing Ca_V_1.3_L_Δ116 channels failed to reconstitute Venus fluorescence, similar to what we observed for the full length Ca_V_1.3_L_ channel (Figure 8)." Experiment should be done with 20 mM Ca^2+^ to mimic conditions used for Ca_V_1.3_S_ in Figure 5 and 6. The failure to observe Ca_V_1.3_L_Δ116 reconstitution of Venus fluorescence could be simply due to the 10-fold less extracellular Ca^2+^ used in this experiment. If so, this would change the conclusion reached for this experiment.* The reviewer raises an important point about the possibility that increasing extracellular Ca^2+^ concentration to 20 mM could induce Ca_V_1.3_L_Δ116 coupling. We had performed the experiments using 20 mM Ca^2+^ previously and did not observe any increase in Venus reconstitution for the Ca_V_1.3_L_Δ116.

The reason we did not include those data in the revised manuscript is because if removing the C116 fragment were to generate a channel capable of coupling, we should be able to discern Venus reconstitution even at 2 mM Ca^2+^, as was true for the Ca_V_1.3_S_ (Figure 8 and Figure 5—figure supplement 1). Nonetheless, for clarity we now added a new supplemental figure to the revised manuscript displaying these data (Figure 8—figure supplement 1). The new Figure is referred to in the last paragraph of the subsection “Deletion of DCRD is not sufficient to allow coupling in Ca_V_1.3_L_ channels”.

*5) Discussion: "An intriguing observation in our study is that physical Ca_V_1.3_S_ channel coupling (assessed by Venus reconstitution) and the normalized conductance of I_Ca_ (i.e., G/G_max_) have similar voltage dependencies. We propose that this is a consequence of two factors. First, Venus fluorescence (i.e., reconstitution) and Ca_V_1.3_S_ conductance have similar sigmoidal relationships suggesting that Ca_V_1.3_S_ coupling is primarily determined by the open probability of these channels. This is consistent with our model, as the extent of Venus reconstitution depends on the local calcium concentration produced by the coordinated opening of clustered channels and less on magnitude of the flux through individual channels. The second factor contributing to the shared voltage dependencies is the irreversible nature of the Venus reconstitution, resulting in an increased number of fluorescent proteins with each successive depolarization as the open probability of Ca_V_1.3_S_ channels increases." These lines should be removed as they represent an over-interpretation of the data. As the authors note in their second point, the sigmoidal nature of the Venus reconstitution dependence on voltage most likely reflects the irreversible nature of the Venus reconstitution such that fluorescence accumulates with successive depolarizations. I suspect if the experiments were done in reverse (i.e., test pulses stepping down from +60 mV to -60 mV) that the relationship would also show a sigmoidal relationship but with the fluorescence lower at the higher voltages where channel Po is higher. This thought experiment suggests the idea that "…Ca_V_1.3_S_ coupling is primarily determined by the open probability of these channels…" is an oversimplification.* We agree that the sigmoidal nature of the Venus reconstitution dependence on voltage is likely to be a consequence of the irreversibility of Venus reconstitution as we have stated. Accordingly, we removed the discussion of the relationship between physical Ca_V_1.3_S_ channel coupling and the *P_o_* of clustered channels. We have also moved our explanation of Venus reconstitution irreversibility to the Results section, since we consider this information essential in following the inferences we draw from our data (subsection “Ca_V_1.3_S_ channels couple under membrane depolarization in a Ca^2+^-dependent manner”, second paragraph).

*6) Please include the procedures for single-channel recordings in the Methods. The single channel traces appear to have quite long openings (particularly for the Ba^2+^ traces; Figure 1—figure supplement 1). Was an L-type channel agonist included for these experiments? If so this should be explicitly stated in the Methods.*

We thank the reviewer for alerting us to this omission. The following description of our single-channel recording protocol has been added to the revised Methods section:

“Single-channel currents (*i_Ca_*) were recorded from tsA-201 using the cell-attached configuration. Cells were superfused with a high K^+^ solution to fix the membrane potential at ∼0 mV. […] The single-channel event-detection algorithm of pClamp 10.2 was used to measure single-channel opening amplitudes and the all-points histograms were constructed using Prism 5.0a software (GraphPad software Inc. La Jolla, CA)”

[Editors’ note: the author responses to the previous round of peer review follow.]

We thank the reviewers and editors for their thoughtful comments on our work and for giving us the opportunity to submit a revised version of the manuscript. Indeed, we are particularly grateful for alerting us to the mutations contained in the plasmids we used to generate the Ca_V_1.3_L_ and Ca_V_1.3_S_ channels that we were completely unaware of. As a consequence, we have repeated virtually all of our experiments and analyses using the corrected versions of the channels. The new manuscript contains a total of 16 new sets of experiments that address the reviewers comments.

The paper is re-submitted nearly nine months after receiving the first round of reviews because the list of necessary experiments was long. That said, we believe the new data were important and think that they have significantly strengthened and improved the paper.

Reviewer #1:

*1) Single channel traces with Ca^2+^ as charge carrier show no signature of Ca^2+^-dependent inactivation despite this being so prominent in whole-cell currents (Figure 1). Why is this? The authors should provide ensemble averages of the single channel data to increase confidence that the unitary currents actually correspond to Ca_V_1.3.* We thank the reviewer for bringing up this important point. We have added new single channel recordings from cells expressing the corrected forms of Ca_V_1.3_L_ and Ca_V_1.3_S_ channels in the revised manuscript (Figure 1). As requested, we also generated ensemble currents from these recordings (see new Figure 1). Ca_V_1.3_S_ and Ca_V_1.3_L_ currents were rapidly activated by depolarization and then inactivated likely due to Ca^2+^ and voltage-dependent mechanisms.

*2) A central point of the paper is that Ca_V_1.3s but not Ca_V_1.3_L_ displays the phenomenon of Ca^2+^-induced channel coupling and facilitation. Ultimately, this is attributed the distal C-terminus competing away CaM binding to the channel. However, the Ca_V_1.3_L_ used in the paper demonstrates quite a robust CDI (e.g. Figure 1), indicating that CaM is bound to the channel. The comparable CDI between Ca_V_1.3s and Ca_V_1.3_L_ observed in the paper is likely due to the use of the rat clones which are known to display similar CDI at ambient CaM concentrations (Yang et al., 2006, J. Neurosci. 26:10677-89). This is because the rat Ca_V_1.3_L_ has a point mutation relative to the human clone, which weakens its affinity for the CaM-binding domain such that ambient CaM is sufficient to (by competition) access its site on the channel (Liu et al., 2010, Nature, 463:968-973). Given this, it is unclear why Ca_V_1.3_L_ would lack the Ca^2+^-dependent facilitation described here. At the very least the interpretation that this due to CaM being excluded from these channels is not supported?* We thank Reviewers 1 and 2 for noting that using the mutated Ca_V_1.3_L_ could confound the central conclusion of our study. As requested, we repeated all our experiments using the corrected version of the Ca_V_1.3_L_ channel to be completely sure that the lack of coupling of the Ca_V_1.3_L_ was not an effect of the mutations contained in the construct. Virtually all the results were identical to what we have seen with the previous construct containing the substitution of V by A in position 2075 (Ca_V_1.3_L(V2075A)_) and they are summarized below:

a) As expected, the currents produced by Ca_V_1.3_L_ channels inactivated at a slower rate and had a smaller CDI when compared to Ca_V_1.3_S_ channels (see new Figure 1);

b) Macroscopic Ca_V_1.3_L_ currents were not enhanced by Ca^2+^ (see new Figure 1);

c) Ca_V_1.3_L_ channels showed a lower sparklet coupling coefficient compared to Ca_V_1.3_S_. (see new Figure 2);

d) When expressed in tsA-201 cells, Ca_V_1.3_L_ channels organized in clusters with a similar average size area compared to Ca_V_1.3_S_ (see new Figure 3—figure supplement 1);

e) The Ca_V_1.3_L_ channels did not exhibit any current facilitation when they were forced to couple using the light-activated cryptochrome system (see new Figure 4);

f) When fused to the split Venus system, the Ca_V_1.3_L_ channels failed to reconstitute Venus fluorescence after depolarization (see new Figure 8).

The only difference observed between Ca_V_1.3_L_ and the previously used Ca_V_1.3_L(V2075A)_ was that, after removing the DCRD domain, the Ca_V_1.3_L_∆116 channels were still unable to couple (see new Figure 8). The answer to the question of why the Ca_V_1.3_L(V2075A)_ is able to couple when the DCRD is removed is still unclear to us. Although interesting, the study of the implication of this difference in the molecular mechanisms by which the channel coupling is translated to longer channel openings is a topic that we will pursue in future studies.

For the present study, we restricted our discussion to the fact that as Ca_V_1.3_L_∆116 channels still have a C-terminus (396 aa) that is considerably longer than that of Ca_V_1.3_S_ channels, it is possible that another domain inside this region might be responsible for occluding the binding of CaM to the pre-IQ domain, and preventing Ca_V_1.3_L_∆116 channel coupling. This discussion was added in the last paragraph of the subsection “Deletion of DCRD is not sufficient to allow coupling in Ca_V_1.3_L_ channels”.

In addition, we performed experiments with the corrected version of the Ca_V_1.3_S_ where a single point substitution of a glycine by a serine at position 244 is corrected. We found that both channels have similar voltage-dependencies of activation and rate of activation (see new Figure 5—figure supplement 2). The corrected Ca_V_1.3_S_ channels were also capable of undergoing Ca^2+^-dependent dimerization (split Venus reconstitution) and functional coupling (Figure 5—figure supplement 2–2F). On the basis of these data, we concluded that the G244S substitution in the plasmid #26576 is functionally silent with respect to activation, rate of activation as well as the ability of Ca_V_1.3_S_ channels to undergo allosteric interactions. Therefore, we decided to clarify in the Methods section of the manuscript (subsection “Plasmid constructs and tsA-201 cell transfection”), that both plasmids were used to express and design the functional Ca_V_1.3_S_ constructs reported in this study.

*3) What is the kinetics of the facilitation? Figure 1 suggest it occurs immediately upon channel activation within 1 ms. This seems very fast for the inter-molecular binding mechanism proposed in Figure 10. Also, how fast does the system reset after closing the channel?* We certainly agree with the reviewer that determining the kinetics of the coupling of Ca_V_1.3_S_ channels is an interesting question. We have shown recently for the Ca_V_1.2 channels that it can be done using FRET (Dixon et al. *eLife* 2015). However, we have not yet done those experiments for the Ca_V_1.3_S_ channels and would like to defer that work for inclusion in our subsequent study.

*4) Does CaM_1234_ prevent the facilitation observed with Ca_V_1.3s in whole-cell currents (i.e. what does Figure 1 look like with CaM_1234_)?* This is an interesting question. We presented evidence showing that CaM_1234_ prevents the increase in coupling coefficient and the increase in sparklet activity (n*Ps*) in response to depolarization, compared to the 7-fold increase seen in control conditions (Figure 6). We consider that the reduction of the n*Ps* is a good piece of evidence that CaM_1234_ prevents the coupling and the facilitation of Ca_V_1.3_S_ channels.

*5) Light-induced dimerization experiments in Figure 4. Seems to me that these experiments should be done with Ba^2+^ and not Ca^2+^ as charge carrier? If idea is the dimerization is sufficient to induce facilitation, having Ca^2+^ also present complicates the interpretation.* We appreciate the reviewer's comment, as it made us cognizant of the fact that the rationale for these experiments and their interpretation were not clearly described in the original version of the manuscript. We now clarify in the subsection “Physical interaction between Ca_V_1.3_S_ channels induces *I_Ca_* facilitation”, that we do not believe dimerization itself is sufficient to induce facilitation. We have adduced many lines of evidence showing that Ca^2+^ and CaM are critically required for physical and functional channel coupling of Ca_V_1.3_S_ channels. Thus we propose that fusing adjoined Ca_V_1.3_S_ channels at the tip of their C-termini with the CIBN-CRY2 system increases the probability of functional coupling, but is not likely sufficient to induce functional channel coupling.

*6) What is the CaM that actually produces the proposed channel coupling? The ability of CaM_1234_ to prevent facilitation would suggest that apoCaM preassociated with the channel is responsible. However, the effectiveness of MLCKp suggests a soluble CaM. If both of these are true then it is inconsistent with the simple model proposed in Figure 10. The authors need to provide a model that fits all the facts they present.* Thanks for the suggestion to make a more complete model. Our results suggest that both tethered and soluble apoCaM bind the pre-IQ domain to induce channel coupling. We modified the model accordingly (see Figure 11) and stated both possibilities in the figure legend and in the manuscript (subsection “Functional Ca_V_1.3_S_-to-Ca_V_1.3_S_ channel coupling is mediated by physical interactions between Ca^2+^-CaM and the pre-IQ domain“, first paragraph).

*7) Figure 6: Is the increase in firing rate decreased by L-type channel antagonists?*

Determining the effect of L-type antagonists in the increase in firing rate is a good suggestion. We performed additional experiments to answer this question. The application of 10 μM nifedipine in hippocampal neurons, expressing Ca_V_1.3_S-_CIBN/CRY2, not only eliminated the light-induced increase in firing rate but decreased significantly the spontaneous firing (Figure 10). This highlights the importance of low-voltage-activated L-type channels in the firing pattern of hippocampal neurons.

Reviewer #2:

*Two major points need to be considered to fully support the conclusions drawn from the experiments presented: 1) The authors describe the use of the Ca_V_1.3 alpha1 rat constructs Addgene plasmid 26577 (long) and 26576 (short). This selection is very unfortunate. Both constructs contain a number of mutations originally introduced by cloning the full-length alpha1-subunits. Addgene (meanwhile?) also provides the corrected versions (plasmids 49332 and 49333).*

*Unfortunately, the mutations are not functionally silent. The Yue group (Liu et al. Cell, 2010) reported that one of the mutations (Val to Ala in ICDI) disrupts the function of a distal modulatory domain (ICDI, also termed DCRD; a finding later confirmed by others: Huang et al. Mol Pharmacol, 2013; Lieb et al., Biophys J 2014). Since this domain competitively tunes the interaction of calmodulin with its upstream binding sites (pre-IQ, IQ) the long mutant Ca_V_1.3 channel behaves functionally more like short splice variants (that lack ICDI due to splicing), with higher CDI and higher open probability, properties that are crucial for the interpretation of the author's data. The authors should have been aware of this because the difference between Ca^2+^ current inactivation of their Ca_V_1.3_L_ and Ca_V_1.3_S_ and between Ca_V_1.3_L_ and Ca_V_1.3delta116 was much smaller than expected between rat (and human) Ca_V_1.3_L_ and Ca_V_1.3_S_ variants (shown in many publications e.g. by the groups of Yue, Soong, and Striessnig).*

*To get around this problem, at least some key experiments affected by the use of mutant constructs should be repeated with their corrected counterparts, in particular those involving Ca^2+^ signals generated by the long isoform (Figure 1., Figure 2, Figure 4). This is justified due to the lower open probability and less pronounced CDI expected for the corrected (wildtype) rat Ca_V_1.3_L_ construct. Although there are also good arguments to repeat all other experiments with the correct constructs, the authors should be able to provide convincing arguments that this is not necessary, e.g. by discussing that CDI does not seem to be a determinant for coupling (Figure 7).*

*2) Evidence for the specificity of staining of endogenous Ca_V_1.3 alpha1 subunits in hippocampal neurons should be provided. So far no Ca_V_1.3 antibody staining has been reported in hippocampal neurons or brain sections that was shown to be absent in Ca_V_1.3 knockout controls. I would therefore not be surprised if this is also true for their antibody. If there is no evidence for such convincing specificity, the authors should either remove data showing staining of native channels or clearly state this caveat in their paper. Obviously, they cannot be blamed for the fact that no anti-Ca_V_1.3 antibody with proven specificity exists. In any case, they should show the typical somatodendritic clusters also on spines of their Ca_V_1.3 constructs transfected in hippocampal cultures using high resolution light microscopy as has been shown in previous work (e.g. Jenkins et al., J Neurosci, 2010).*

We agree with the reviewer that, so far, no anti-Ca_V_1.3 antibody with proven specificity in KO controls exist and thank for the suggestion to clarify this caveat in the manuscript. As suggested, this limitation in our study is now noted in the revised manuscript (subsection “Ca_V_1.3 channels in hippocampal neurons aggregate in dense clusters”, first paragraph). In addition, we now clarify in the manuscript that as we were not able to test the specificity of the antibody on Ca_V_1.3-KO neurons, we pursued our analyses of channel clustering using a heterologous expression of Ca_V_1.3 channels in tsA-201 cells. The specificity of the antibody in tsA-201 cells was tested by immunostaining untransfected and Ca_V_1.3_S_-transfected cells. No evidence of staining was observed in the untransfected cells (see new Figure 3—figure supplement 1).

We also consider that our results using the step-photobleaching of expressed Ca_V_1.3_S_-mGFP is a strong piece of evidence that these channels organize in clusters in hippocampal neurons. In addition, the average number of channels per cluster in both hippocampal neurons and tsA-201 cells correlates with the mean areas calculated from the super-resolution data. Finally, we were able to see an increase in the cluster number when we compared untransfected and Ca_V_1.3_S_VC/VN transfected hippocampal neurons (Figure 10—figure supplement 1).

*3) In the subsection “Coupling of Ca_V_1.3_S_ channels increases the firing rate of hippocampal neurons”: According to Moosmang et al. (2005; PMID: 16251435), Ca_V_1.3 current components are minimal in hippocampal neurons and L-type currents seem to comprise only a small fraction of total voltage-gated calcium current. So how do the authors know their sparklets are produced by L-type currents? If so, they should vanish after treatment with a dihydropyridine Ca^2+^ channel blocker. This important control appears to be missing in the manuscript. They should also consider that there is no way to distinguish between Ca_V_1.3 and Ca_V_1.2 sparklets (not even by membrane voltage as stated in their manuscript).*

We performed the control experiment suggested by the reviewer. The new Figure 9 includes data showing that sparklet site activity was decreased by application of 300 nM of the dihydropyridine antagonist isradipine and completely eliminated when the concentration of the drug was increased to 10 µM (Figure 9). This is consistent with the hypothesis that sparklets in hippocampal neurons were produced by L-type calcium channels. Both Ca_V_1.2 and Ca_V_1.3 channels are expressed in hippocampal neurons (Hell et al., 1993) and although it is impossible to distinguish between these two L-type Ca^2+^ channels using either electrophysiological or pharmacological methods, it has been shown that Ca_V_1.3 channels have a reduced sensitivity to dihydropyridines compared to Ca_V_1.2 channels (Lipscombe et al., 2004; Xu and Lipscombe, 2001). A previous study by Koschak et al. found that 100% of Ca_V_1.2 channels but only ~60% of Ca_V_1.3 channels are inhibited by 300 nM isradipine (Koschak et al., 2001), given this, it is reasonable to assume that any sparklet remaining after application of 300 nM isradipine is more likely to be generated by Ca_V_1.3 channels than Ca_V_1.2. This discussion was added to the manuscript (subsection “Coupling of Ca_V_1.3_S_ channels increases the firing rate of hippocampal neuron”, first paragraph).